# Delivery of Protein Kinase A by CRISPRMAX and Its Effects on Breast Cancer Stem-Like Properties

**DOI:** 10.3390/pharmaceutics13010011

**Published:** 2020-12-23

**Authors:** Jun-Nian Zhou, Tzu-Chen Rautio, Chang Liu, Xiao-Yu Xu, Dong-Qing Wang, Yong Guo, John Eriksson, Hongbo Zhang

**Affiliations:** 1Experimental Hematology and Biochemistry Lab, Beijing Institute of Radiation Medicine, Beijing 100850, China; zhoujunnian@scrm.org.cn; 2Stem Cell and Regenerative Medicine Lab, Institute of Health Service and Transfusion Medicine, Beijing 100850, China; 3Pharmaceutical Sciences Laboratory, Åbo Akademi University, 20520 Turku, Finland; tzu-chen.t.rautio@utu.fi (T.-C.R.); chang.liu@abo.fi (C.L.); xiaoyu.xu@abo.fi (X.-Y.X.); 4Turku Bioscience Center, University of Turku, 20520 Turku, Finland; 5Turku Bioscience Center, Åbo Akademi University, 20520 Turku, Finland; 6Institute of Biomedicine, Faculty of Medicine, University of Turku, 20520 Turku, Finland; 7Department of Radiology, Affiliated Hospital of Jiangsu University, Jiangsu University, Zhenjiang 212001, China; wangdongqing71@ujs.edu.cn; 8Department of Endocrinology, Key Laboratory of National Health and Family Planning Commission for Male Reproductive Health, National Research Institute for Family Planning, Beijing 100081, China

**Keywords:** CRISPRMAX, drug delivery, protein kinase A, chemoresistance, cancer stem cells, epithelial-mesenchymal transition

## Abstract

Protein kinase A (PKA) activation has recently been reported to inhibit epithelial-mesenchymal transition (EMT) and cancer stem cell (CSC) ability, which is considered to be responsible for chemoresistance and tumor recurrence in patients. While current studies mainly focus on gene manipulation of the EMT process, the direct delivery of PKA enzymes to cancer cells has never been investigated. Here, we utilize the commercial Lipofectamine CRISPRMAX reagent to directly deliver PKAs to breast cancer cells and evaluate its effects on EMT regulation. We optimized the delivery parameters with fluorescent-labeled bovine serum albumin, and successfully delivered fluorescent PKAs through CRISPRMAX into breast cancer cells. Then, we evaluated the biological effects by immunofluorescence, flow cytometry, mammosphere assay, and chemoresistance assay. Our data showed the expression of EMT-related markers, α-smooth muscle actin and N-cadherin, was downregulated after CRISPRMAX-PKA treatment. Although the CD44^+^/CD24^−^ population did not change considerably, the size of mammospheres significantly decreased. In paclitaxel and doxorubicin chemoresistance assays, we noticed PKA delivery significantly inhibited paclitaxel resistance rather than doxorubicin resistance. Taken together, these results suggest our direct enzyme delivery can be a potential strategy for inhibiting EMT/CSC-associated traits, providing a safer approach and having more clinical translational efficacy than gene manipulation. This strategy will also facilitate the direct testing of other target enzymes/proteins on their biological functions.

## 1. Introduction

The cancer stem cell (CSC) theory has been proposed as one of the viable explanations for the resistance of cancer cells to conventional cancer therapies, including chemotherapy and radiotherapy in the clinic [1]. Emerging studies demonstrate that epithelial-to-mesenchymal transition (EMT) plays a crucial role in the dedifferentiation of differentiated cancer cells into CSCs [2,3]. Many key EMT regulators, including transcription factor Snail family transcriptional repressor 1 (SNAIL1), Twist family BHLH transcription factor 1 (TWIST), zinc finger E-Box binding homeobox 1 (ZEB1), and microRNAs (miRNAs), such as miR-200 family, let-7, have been revealed and have shown promise in inhibition of the EMT process in studies [4]. Therefore, targeting EMT-induced CSCs or reversing EMT has become an attractive strategy for inhibiting CSCs and CSC-associated chemoresistance.

As a kind of biological macromolecule, enzymes are one of the main targets in the development of many disease-associated drugs in cancers. Many small-molecule compounds have been developed as inhibitors or agonists of enzyme activity and are widely used in antitumor research and treatment. On the other hand, enzymes can also be used directly as medicine. For example, therapeutic enzymes used in replacement therapy for genetic diseases. Adagen1 (pegadamase bovine) was the first therapeutic enzyme approved by FDA under the Orphan Drug Act in 1990. Compared with other therapeutics, enzymes have two important properties as drugs: they specifically bind to the targets, have catalytic properties, and can catalyze many target molecules into target products [5]. Based on previous studies mentioned above, we hypothesized that introducing a functional enzyme that inhibits EMT/CSCs as a candidate drug into cancer cells could inhibit EMT/CSC-associated properties, leading to a catalytic specificity and lower cytotoxicity. Most enzymes have been reported to be overexpressed in the EMT process [6,7,8], but we cannot intervene in the EMT process by directly delivering enzymes into cells. Recently, Pattabiraman et al. reported that induction of protein kinase A (PKA) can evoke mesenchymal human mammary epithelial cells that undergo mesenchymal-to-epithelial transition (MET) [9], an inverse process of EMT, which promotes the differentiation of CSCs, causing the loss of tumor-initiating capacity. This indicates PKA may be an ideal target for our direct enzyme transfer platform.

Because PKA enzyme needs to function inside the cell, we need a carrier to transfer it into the cell while maintaining the activity of the enzyme. In recent times, rapid developments in nanoparticle-based drug delivery systems have led to improvements in the therapeutic efficacy of cancer treatment [10,11]. Among these particles, lipid nanoparticles (LNPs) are the most common and well-investigated nanocarriers for targeted drug delivery. They have many advantages, such as flexible physicochemical and biophysical properties, easy operation for different delivery requirements, high efficiency of entering cells and tissues, and reduced system toxicity [12], etc. In 1995, the first clinically approved nanomedicine for the treatment of cancer was liposomal doxorubicin (Doxil) [13], which was expected to reduce the cardiotoxicity of doxorubicin (DOX). The poorly water-soluble chemotherapy drug paclitaxel has also been prepared as liposomal paclitaxel in order to improve its efficiency in targeting tissues [14]. There are also other chemotherapy drugs encapsulated in LNPs [15], and some have been approved for clinical trials. On this basis, biomolecules, such as nucleic acid molecules, are also used for delivery by LNPs, such as miRNA and siRNA that can regulate the expression of target proteins [16,17]. They are not only widely used in life science-related laboratories to study the function of genes but also in the field of tumor treatment research [18]. The focus for this study is the biological understanding of a direct delivery of PKA enzymes into the cancer cells. PKA has the function of inhibiting epithelial-to-mesenchymal transition and cancer stem cells, thus it has great potential against resistance to cancer treatment. In this work, we utilized a commercial ready-to-use product, CRISPRMAX (Lipofectamine CRISPRMAX Cas9 transfection reagent), which is the first optimized lipid nanoparticle transfection reagent for CRISPR-Cas9 protein delivery [19,20,21]. Since this commercial reagent has been well established, the results obtained by using this commercial kit will provide a cornerstone of enzyme delivery strategy for broader biological community, and these results will be easily repeated by other laboratories, even they have no material science background. This will also be a practically handled method for clinical doctors in future clinical applications.

In fact, there are many other functionalized nanosystems that can be used to deliver proteins/enzymes. For example, in our research group, we have developed different platforms for protein delivery, including: (a) liposomes [22], (b) polymer nanoparticles [23], (c) polymersomes [24], (d) mesoporous nanoparticles [25], (e) metal-organic framework with biomineralization [26], (f) DNA nanoparticles [27,28], (g) emulsions [29], etc. However, in the end, we selected the commercial CRISPRMAX, because CRISPRMAX has the following advantages. First, this reagent has low toxicity to cells, allowing us to focus on evaluating the biological effects of PKA enzymes on cells, including epithelial-mesenchymal transition, cancer stem cells, and chemoresistance. Second, the experimental conditions for this reagent have been optimized; there is a standard protocol for CRISPRMAX-enzyme preparation and cell transfection, which is involved in cell seeding density, reagent usage, the time of each step, and the precautions for each step. Third, such reagent is an ideal delivery solution in 96-well format plates for high-throughput analysis. These advantages make us easily focus on the biological function of a direct delivery of PKA enzymes into the cancer cells rather than nanocarriers themselves, and provide a good paradigm of potential translational application in the clinic.

Therefore, Lipofectamine CRISPRMAX reagent was selected as a model nanocarrier for investigating the establishment of direct transfer of PKAs into breast cancer cells. We used fluorescent-labeled bovine serum albumin (BSA) as a positive control to optimize relevant transfection parameters, and proved that PKA enzymes through CRISPRMAX can be introduced into breast cancer cells. The biological effects of CRISPRMAX-PKA complexes were further examined through functional experiments, including chemoresistance assay, flow cytometry (FCM), and mammosphere assay, to test whether the CSC- and/or EMT-related properties were repressed through the biotherapeutic effects of the complexes. We observed that the expression of EMT-related markers, α-smooth muscle actin (α-SMA) and N-cadherin (N-cad), was downregulated. Although the CD44^+^/CD24^−^ population did not change significantly, in another functional test of CSCs, we found that the size of mammospheres (MSs) was significantly downregulated. In the chemoresistance assay, we observed a drug-dependent result, that is, PKA delivery has a significant inhibitory effect to paclitaxel (PTX) resistance, but there is no effect against DOX. Interestingly, this inhibitory effect is order dependent, that is, the inhibitory effect can be exerted only in the cells that were treated with PKA before PTX administration. This indicates that the role of PKA enzymes in chemotherapy drug resistance may be drug dependent. Taken together, as a proof-of-concept, we confirmed the direct enzyme delivery as a potential strategy for inhibiting EMT/CSC-associated chemoresistance. This strategy will also facilitate the screening of target enzymes/proteins for the construction of a new drug delivery system or complex nanoparticles.

## 2. Materials and Methods

### 2.1. Cell Culture

Human breast cancer MDA-MB-231 cells and MCF-7 cells were cultured with Dulbecco’s Modified Eagle’s medium (DMEM, Lonza, Walkersville, MD, USA), which contained 10% fetal bovine serum (FBS, Gibco, Grand Island, NY, USA), 0.5% penicillin/streptomycin (Pen/Strep, Gibco), and 1% L-Glutamine (Gibco). Human breast cancer BT-549 cells and Hs-578T cells were grown in ATCC-modified RPMI 1640 medium (Gibco), which contained 10% FBS, 0.5% Pen/Strep, and 1% L-glutamine. Human normal breast cells MCF-10A were cultured in DMEM/F12 medium (Gibco) containing 5% FBS, D-(+)-Glucose (4.5 g/L, Sigma, St. Louis, MO, USA), human insulin (10 μg/mL, Sigma), cholera toxin (Sigma, 0.1 μg/mL), hydrocortisone (0.5 μg/mL, Sigma), epidermal growth factor (EGF, 20 ng/mL, Peprotech, London, UK), and 1% L-glutamine. All the cells were cultured in a humidified incubator with 5% CO_2_ at 37 °C. MCF-7 cells, BT-549 cells, Hs-578T, and MCF-10A cells were kindly provided by Prof. Jukka Westermarck (Turku Bioscience, Turku, Finland).

### 2.2. Enzyme Delivery by Lipofectamine CRISPRMAX Reagent

Lipofectamine CRISPRMAX reagent (Invitrogen, Carlsbad, CA, USA) was used for delivering BSA and PKA enzymes in this study. All procedures were followed according to the manufacturer’s protocol. Breast cancer cells and MCF-10A cells were seeded in 96-well plates (5000 cells per well, except MDA-MB-231 cells were seeded at the density of 2000 cells per well) one day before transfection. The seeded plates were incubated overnight; the cells were grown at 50–70% confluence in 96-well plates for transfection. On the day of transfection, the culture medium of each well in the plates was replaced with 100 μL of Opti-MEM (Invitrogen) before performing CRISPRMAX-based enzyme delivery. As for the preparation of enzymes-loaded CRISPRMAX, tube 1 contained 5 μL of Opti-MEM medium, 0.5 μL of PKA enzyme solution (500 ng per well, 1 μg/μL of stock solution), and 0.5 μL of Cas9 Plus reagent were added in sequence. The control cells were treated with the same amount of purified BSAs, or Alexa Fluor 680-labeled BSAs (Invitrogen). Tube 2 contained 5 μL of Opti-MEM medium with 0.5 μL of CRISPRMAX reagent. Next, the solution of tube 1 was added to tube 2 within 3 min, and then mixed well; the complex solution was incubated at room temperature. After 5–10 min, 10 μL/well of CRISPRMAX complex solution was added into the cancer cells for 1 day of incubation. Then, the CRISPRMAX-containing medium was replaced with complete growth medium in each well; the transfected cells continued to grow for 1 to 3 days before conducting further experiments.

### 2.3. Characterization of PKAs- and BSAs-Loaded CRISPRMAX Complexes

The morphology of the enzymes-loaded CRISPRMAX complexes was observed by means of transmission electron microscopy (TEM) and confocal laser scanning microscopy (CLSM). For TEM analysis, fresh CRISPRMAX-BSA complexes, CRISPRMAX-PKA complexes, and null CRISPRMAX complexes were prepared, respectively, according to the protocol mentioned in Section 2.2. The grids used for the electron microscope samples were immersed in the complex solution for each setup for a few seconds, then taken out, and placed on an absorbent paper for drying overnight. The grids were observed under TEM (LVEM 5, Delong Instruments, Brno, Czech Republic). For CLSM analysis, fluorescent BSAs (Alexa Fluor 680-labeled BSAs) were used to prepare fresh CRISPRMAX-BSA-Alexa 680 complexes; the same concentration of BSA-Alexa 680 solution without CRISPRMAX was used as a control group. The solution of the complexes or the control group was dripped onto the glass slides, and then these samples were covered with cover glasses; fluorescent particles were observed under CLSM (Zeiss, LSM880 with Airyscan, Carl Zeiss AG, Oberkochen, Germany). The particle size of CRISPRMAX-PKAs and CRISPRMAX-BSAs was measured with dynamic light scattering using a Zetasizer Nano ZS (Malvern Instruments Ltd., Malvern, Worcs, UK). The surface charge of the complexes (zeta potential) was measured using a Zetasizer Nano ZS with disposable folded capillary cells (DTS1070, Malvern, Worcs, UK).

### 2.4. Encapsulation Efficiency

Fluorescent-labeled BSAs (BSA-Alexa 680) were used to calculate the encapsulation efficiency of CRISPRMAX. Briefly, fresh CRISPRMAX-BSA-Alexa 680 complexes were prepared, according to the protocol mentioned in Section 2.2. In the meantime, the same amount of BSA-Alexa 680 without CRISPRMAX was prepared in the same volume of solution as positive control groups, of which the fluorescence intensity represented the total amount of BSA-Alexa 680 (V_t_). Then, the complex solution was added into the upper chamber of the centrifugal filter device (100K, Merck Millipore, Billerica, MA, USA), followed by centrifugation at 13,000 rpm for 5 min. The molecular weight of CRISPRMAX and CRISPRMAX-BSA-Alexa 680 complexes is greater than 100 kDa, and the complexes will be trapped in the filter, while the molecular weight of unencapsulated free BSA-Alexa 680 is less than 100 kDa, and will be passed through the filter. After centrifugation, the filtrate was collected. Through this method, the unencapsulated free BSAs in the preparation system were obtained. The fluorescence intensity of these complexes was analyzed with a Varioskan multimode reader, and the fluorescence intensity of BSA-Alexa 680 (V_x_) was obtained. The encapsulation efficiency was calculated based on the following formula:Encapsulation efficiency (%) = (V_t_ − V_x_)/V_t_ × 100%(1)

### 2.5. Release Efficiency

For in vitro release study of CRISPRMAX, fluorescent BSAs (BSA-Alexa 680) were used to calculate the release efficiency of CRISPRMAX. The release properties were measured at the following time points: 0, 1, 2, 4, 8, 12, and 24 h. First, fresh CRISPRMAX-BSA-Alexa 680 complexes were prepared, according to the protocol mentioned in Section 2.2; 1.25 μg of BSA-Alexa 680 in 250 μL of complex solution per tube, each tube for one time point, respectively. Meanwhile, the same amount of BSA-Alexa 680 without CRISPRMAX (1.25 μg of BSA-Alexa 680 in 250 μL of solution without CRISPRMAX per tube, each tube for one time point, respectively) was prepared as positive control groups, of which the fluorescence intensity represented the total amount of BSA-Alexa 680 (V_t_). All samples of the release groups and control groups were put into a shaking water bath at 37 °C, protected from the light. For the experimental setup, the solution of CRISPRMAX-BSA-Alexa 680 complexes was added into the upper chamber of the centrifugal filter device (100K, Merck Millipore), followed by centrifugation at 13,000 rpm for 5 min; the filtrate of the release groups per setup was collected. The fluorescence intensity values of the free BSA-Alexa 680 solution (V_0_, V_1_, V_2_, V_4_, V_8_, V_12_, and V_24_) in the release groups were measured at each time point. The fluorescence intensity values of the positive control groups (V_t0_, V_t1_, V_t2_, V_t4_, V_t8_, V_t12_, and V_t24_) were also measured at each time point. Based on the fluorescence value of the positive controls at each time point, the release percentage was normalized at each time point, and subtracted the proportion of the original unencapsulated free BSA-Alexa 680. For example, the percentage release at the first hour time point was calculated as follows:Percentage release (%) = [(V_1_/V_t1_) − (V_0_/V_t0_)] × 100%(2)

### 2.6. Cellular Uptake Efficiency Analysis of CRISPRMAX-BSA-Alexa Fluor 680

According to the protocol mentioned in Section 2.2, 500 ng of BSA-Alexa Fluor 680 was coated by CRISPRMAX reagent and added into BT-549 cells per well (*n* = 3). The cellular uptake was measured at indicated time points (10 min, 30 min, 1 h, 2 h, 3 h, 4 h, 6 h, and 23 h) after CRISPRMAX-BSA-Alexa 680 complex treatment in cells. In the meantime, the fluorescence intensity value of CRISPRMAX-BSA-Alexa 680 without cells (*n* = 3) was measured at indicated time points. The fluorescence intensity value for each sample was measured at 679 (excitation) and 702 nm (emission) with a Varioskan LUX multimode reader (Thermo Fisher Scientific, Waltham, MA, USA). Cellular uptake efficiency was calculated based on the following formula: I_c_ = the fluorescence intensity value of BSA-Alexa 680 (500 ng) only in 110 μL of OPTI-MEM without CRISPRMAX complex and cells, and I_t_ = the fluorescence intensity value of BSA-Alexa 680 in CRISPRMAX-complex-containing cell transfection supernatant at the indicated time points.Cellular uptake efficiency = (I_c_ − I_t_)/I_c_ × 100%(3)

### 2.7. Immunofluorescence Analysis by Confocal Laser Scanning Microcopy

CRISPRMAX-PKAs- and CRISPRMAX-BSAs-delivered cancer cells were seeded in confocal dishes (Thermo Fisher Scientific) and continued to grow for 3 to 5 days after delivery. Cell fixation was performed using 4% paraformaldehyde (PFA, Sigma) in the confocal dishes for 30 min at room temperature. Then, 0.1% triton X-100 (Sigma) was used to permeabilize the cell membrane of the samples for 5–10 min, followed by blocking with 10% goat serum (Life Technologies, Auckland, New Zealand) for 30 min at room temperature without PBS washing after discarding. The primary antibodies were directly incubated with the cells according to the manufacturer’s protocol. For EMT markers expression analysis, mouse anti-human E-cad antibody (1:50, Santa Cruz Biotechnology, Santa Cruz, CA, USA), mouse anti-human N-cadherin antibody (1:50, Santa Cruz Biotechnology), and mouse anti-human α-SMA antibody (1:100, Santa Cruz Biotechnology) were diluted in PBS and incubated with the cells overnight at 4 °C, followed by TRITC-labeled goat anti-mouse secondary antibodies incubation for 30 min at room temperature. For PKA expression analysis, rabbit anti-human/bovine PKA antibodies (1:300, Invitrogen) were incubated with the cells overnight at 4 °C, followed by TRITC-labeled goat anti-rabbit secondary antibodies incubation for 1 h at room temperature. All prepared samples were rinsed with PBS three times before nuclei staining. To visualize the cell nuclei of these samples, the samples were treated with 5 μg/mL of 4′,6-diamidino-2-phenylindole (DAPI, Life Technologies) solution for 5 min at room temperature; the stained samples were observed using confocal laser scanning microcopy (CLSM) (Zeiss, LSM880).

### 2.8. Flow Cytometry

For analysis of CD44 and CD24, the flow cytometric samples were prepared according to the manufacturer’s protocol (Invitrogen). Briefly, the trypsinized single cell suspension was washed with PBS twice and centrifuged for 8 min at 330 g. For each test (100 μL of PBS), 0.15 μL of mouse anti-human APC-labeled CD44 monoclonal antibody (eBioscience, San Diego, CA, USA) and 5 μL of mouse anti-human PE-labeled CD24 monoclonal antibody were incubated together in an Eppendorf tube for 15 min at 4 °C, which was packed with aluminum lamination foils to protect from light exposure. Similar to the preparation of CD44 and CD24 markers, the amount of isotype controls (APC-labeled mouse IgG and PE-labeled mouse IgG) were prepared, that was, 0.3 μL of APC-IgG (eBioscience) and 1.25 μL of PE-IgG (eBioscience) in a total volume of 100 μL of PBS. After antibody incubation, the cells were washed with PBS 2–3 times and centrifuged for 8 min at 330 g. Then, the samples were resuspended in 500 μL of PBS for flow cytometric analysis with a BD LSR Fortessa analyzer.

### 2.9. Mammosphere Formation Assay

Mammosphere assay has been used for identification of tumor spherical colonies, CSCs, also known as mammospheres (MSs) [16,30,31]. The single cell suspensions were obtained by trypsinizing the breast cancer cells that were transfected with CRISPRMAX-PKAs or CRISPRMAX-BSAs. The same number of the cancer cells from each group was grown in ultra-low attachment plates (Sigma) containing serum-free DMEM/F12 (Lonza), B27 (1:50, Gibco), N2 (1:100, Gibco), 2 μg/mL of human insulin solution (Sigma), 4 μg/mL of heparin (Sigma), human epidermal growth factor (EGF, Peprotech), and human basic fibroblast growth factor (bFGF, Peprotech). The MSs were observed and imaged under a bright-field microscope one week later. Diameter measurement of MSs was analyzed by ImageJ, and the size of MSs greater than 70 μm in diameter was counted.

### 2.10. Chemoresistance Assay

The effects of PKA-loaded CRISPRMAX complexes in breast cancer cells and normal breast cells on the chemoresistance of PTX (Arisun ChemPharm, Xi’an, China) and DOX (Arisun ChemPharm) were determined by WST-1 cell viability assay [26]. Breast cancer cells and MCF-10A cells were seeded in 96-well plates (5000 cells per well, except MDA-MB-231 cells were seeded at the density of 2000 cells per well, three parallel wells for each concentration) in complete growth medium containing 5% FBS, and cultured in the cell incubator overnight. Then, each well of the medium was replaced with fresh Opti-MEM medium containing CRISPRMAX-PKAs or CRISPRMAX-BSAs complexes. After delivering the complexes, PTX or DOX at the indicated concentrations was added under different transfection time points (at 0 and 12 h). Transfected cells were treated with DOX at 0, 0.1, 0.5, 1, and 1.5 μg/mL or with PTX at 0, 0.05, 0.1, 0.2, and 0.5 μg/mL. The duration of PTX and DOX treatment was 48 h. For WST-1 assay, according to the manufacturer’s protocol (Roche, Mannheim, BW, Germany), the drug-containing medium was replaced with 10 μL of WST-1 reagent that was dissolved in 100 μL of complete growth medium for each well. After 2 h of incubation in the cell incubator, the absorbance was measured at 440 nm with a Varioskan LUX multimode reader (Thermo Fisher Scientific).

### 2.11. Statistical Analyses

All data were presented as means with standard deviations (SD); *n* = 3 replicates for transfection with PKA and BSA (control) group. Statistical graphs were generated using GraphPad Prism 8 software (GraphPad Software Inc., San Diego, CA, USA) or Origin 6.1 software (OriginLab, Northampton, MA, USA). Quantitative analysis of the mean of fluorescence intensity of PKA was performed by ImageJ. The data of chemoresistance assays were further performed with AUC (area under curve) to compare the effectiveness of drug effect between transfection with PKA and BSA enzymes. The statistical significance of differences between two groups was determined by unpaired two-tailed *t* tests or two-way ANOVA analysis, according to the number of parameters of the data. *p* < 0.05 was considered statistically significant (* *p* < 0.05, ** *p* < 0.01, and *** *p* < 0.001); *p* > 0.05 was considered not significant.

## 3. Results

### 3.1. Characterization of PKAs- and BSA-Loaded CRISPRMAX Complexes

#### 3.1.1. Morphology

The morphology data of the enzymes-loaded CRISPRMAX complexes were collected by means of TEM and CLSM. Under TEM, the different appearances of CRISPRMAX and the encapsulated proteins were clearly shown. CRISPRMAX was shown as a light gray outer outline of the complexes, while the encapsulated BSA or PKA proteins were presented in black (Figure 1a). For null CRISPRMAX, there was only a light gray outline. These results showed that the proteins were successfully coated by CRISPRMAX. For CLSM analysis, fluorescent BSAs were used to prepare fresh CRISPRMAX-BSA-Alexa 680 complexes; the same concentration of BSA-Alexa 680 solution without CRISPRMAX was used as a control group. Fluorescent particles were clearly observed under CLSM; the images in 2-D and 3-D display showed that small red particles were distributed in the solution, while BSA-Alexa 680 in the control group had no significant red particles due to a lack of CRISPRMAX encapsulation (Figure 1b,c), which further indicates BSA-Alexa 680 proteins were successfully coated by CRISPRMAX.

#### 3.1.2. Encapsulation Efficiency and Release Efficiency of CRISPRMAX

For evaluation of the encapsulation capability of CRISPRMAX, fluorescent-labeled BSAs (BSA-Alexa 680) were used as a model to mimic the encapsulation efficiency of CRISPRMAX, since it is very difficult to directly quantify PKA. The same amount in the same volume of BSA-Alexa 680 without CRISPRMAX as well as in CRISPRMAX-BSA-Alexa 680 complexes was used as a positive control to indicate a total of 100% fluorescence value. Our results showed that the encapsulation efficiency using CRISPRMAX to encapsulate the proteins was 76.11% ± 4.320% (Figure 1d). As for the analysis of the release property of CRISPRMAX, our results showed that there were no significant releases of the free BSA-Alexa 680 from CRISPRMAX-BSA-Alexa 680 complexes within the first four hour. A small release (<10%) was detected at the eighth hour (Figure 1e), but no further release was detected thereafter, indicating CRISPRMAX-BSA-Alexa 680 complexes were relatively stable in in vitro.

#### 3.1.3. Determination of Zeta Potential and Particle Size of the Prepared Lipid Nanoparticles

Two of the most important parameters of nanoparticles are particle size and zeta potential [32]. Zeta potential is the overall surface charge of a particle in a particular solution. In this study, 10 μL of CRISPRMAX complexes was added into 100 μL of Opti-MEM, mixed well, followed by a 20-fold dilution in Opti-MEM. Both the size and zeta potential of these samples were measured using a Zetasizer Nano ZS. The mean of the size of PKA, BSA-loaded, and control CRISPRMAX (null) was found to be 349.8 ± 21.65, 344.5 ± 45.54, 345.56 ± 33.30 nm in Opti-MEM medium, respectively. The overall surface charge of PKA, BSA-loaded, and control CRISPRMAX (null) was found to be −8.38 ± 0.337, −7.87 ± 0.0777, −8.33 ± 0.403 in Opti-MEM medium, respectively. There was no significant difference between each group in terms of size and surface charge (Table 1).

### 3.2. Establishment of a CRISPRMAX-Based Enzyme Delivery System

#### 3.2.1. Establishment of a CRISPRMAX-Based Protein Delivery System Using Alexa 680-Labeled BSA as a Model Protein

In general, most enzymes are sensitive to fluorescent conjugation reactions, easily losing their activities due to the complicated reaction processes; thus, fluorescent BSAs are used as an indicator for confirmation of our established enzyme delivery system that utilizes CRISPRMAX as a delivery platform. To prove this, flow cytometric and immunofluorescence analyses were performed for testing whether BSA-Alexa Fluor 680 could be guided through this transfection system. The flow cytometric histogram demonstrated two distinguished populations, the negative control population transfected with ddH_2_O through CRISPRMAX (shown in blue) and the positive population with fluorescent BSAs loaded in CRISPRMAX (shown in red). Our results indicated fluorescent BSAs were successfully delivered through CRISPRMAX into the breast cancer cells at high efficiency (65.4% at day 4 in BT-549, 65% at day 2 in MDA-MB-231, 78.1% at day 2 in Hs-578T) (Figure 2a,b). Considering the duration of the delivery, BT-549 cells had higher efficiency than MDA-MB-231 and Hs-578T cells. Immunofluorescence analysis by CLSM further suggested that fluorescent BSAs were directly delivered into the cells through CRISPRMAX (Figure 2c). We calculated the delivery efficiency of CRISPRMAX-BSA-Alexa Fluor 680 in the cells. The cellular uptake was measured by a Varioskan Lux microplate reader at indicated time points (10 min, 30 min, 1 h, 2 h, 3 h, 4 h, 6 h, and 23 h) after CRISPRMAX-BSA-Alexa Fluor 680 complex treatment in cells; in the meantime, the fluorescence intensity of the solution containing CRISPRMAX-BSA-Alexa Fluor 680 without cells was measured at the indicated time points. The results showed that (27.9 ± 7.3)% of BSA-Alexa Fluor 680 was delivered into the cells at the 30th minute of transfection, and (44.2 ± 8.1)% of BSA-Alexa Fluor 680 was delivered into the cells at the 23rd hour of transfection (Figure 2d).

#### 3.2.2. PKA Was Upregulated by PKA-Lipid Nanoparticles Delivery

Due to a high sequence similarity of PKA in other organisms (55% identity over 300 amino acids) that contains the conserved serine threonine tyrosine kinase domains, we loaded protein kinase A from bovine heart where PKA is highly expressed. At day 3 of CRISPRMAX-PKAs delivery in BT-549 cells, we performed immunofluorescence analysis using PKA alpha polyclonal antibody that can recognize both human and bovine-derived PKA. Our results showed that PKA was upregulated by PKA-lipid nanoparticles delivery (Figure 3).

Together, these data suggest that we have successfully established a LNPs-based PKA enzyme delivery system.

### 3.3. Analysis of the Effects of PKA-Lipid Nanoparticles Delivery on EMT-Associated Marker Expression and CSCs

#### 3.3.1. Expression Analysis of E-Cadherin, N-Cadherin, and α-SMA after PKA Delivery Using CLSM

Since BT-549 cells have higher efficiency than MDA-MB-231 and Hs-578T cells, herein, we focused on the effects of PKA-lipid nanoparticles delivery on EMT-associated marker expression and CSCs in BT-549 cells. Our results showed that the expression of α-SMA and N-cadherin was decreased in the cells transfected with CRISPRMAX-PKAs compared with the control (CRISPRMAX-BSAs); the expression of E-cadherin was not remarkably different between the PKA and BSA group (Figure 4).

#### 3.3.2. Analysis of the Effects of PKA Enzyme Delivery on CSCs

As mentioned earlier, our immunofluorescence results showed that the expression of EMT-related mesenchymal markers, α-SMA and N-cadherin, was decreased in the breast cancer cells treated with PKA, indicating the suppression of the EMT process at the molecular level (Figure 4). Other functional experiments were also performed to examine the impact on CSCs after introducing PKAs. Flow cytometry detection of changes in cell surface markers is a common method for rapid detection of changes in the CSC population. It is believed that the CD44^+^/CD24^−^ population can enrich the CSC population in breast cancer cells [33]. Therefore, we performed flow cytometry analysis to examine the CD44/CD24 population in four different kinds of breast cancer cells (BT-549, MDA-MB-231, Hs-578T, and MCF-7 cells) 3 to 5 days after CRISPRMAX-PKAs delivery. Compared with the CRISPRMAX-BSAs group, the CD44^+^/CD24^−^ population of the CRISPRMAX-PKAs group did not significantly decrease (Figure 5). Next, we utilized the mammosphere assay, which is considered as one of the most effective evaluation methods of breast CSCs in vitro [34], to evaluate the biological effects of CRISPRMAX-PKAs delivery on breast cancer cells. The size and number of mammospheres are related to the mamosphere-formation ability that represents the stemness of the CSCs in the culture process. Here, we focused on the relatively large size of MSs (diameter > 100 μm). Our results showed that the size of MSs grown in the mammosphere culture system for 7 days was significantly reduced after CRISPRMAX-PKAs delivery (Figure 6). The diameter of CRISPRMAX-PKAs was 134.915 ± 24.123 μm compared with that of CRISPRMAX-BSAs (144.163 ± 31.121 μm) (*p* = 0.0214) (Figure 6b,c). Together, these data suggested that CRISPRMAX-based PKA delivery decreased the expression of mesenchymal markers (α-SMA and N-cadherin) and reduced the mammosphere-forming capacity of breast cancer cells.

### 3.4. Analysis of the Synergy Effects of CRISPRMAX-Based PKA Delivery along with Chemotherapy Drugs

To explore the effects of co-treatment of breast cancer cells with PKA along with the chemotherapy drug (PTX or DOX), and whether the transfection with PKA could increase the chemosensitivity of breast cancer cells, the experimental setups were designed under different transfection times: 0 and 12 h, then PTX or DOX were added after transfection. Data of cell viability were analyzed, and we also performed area under curve (AUC) analysis for measurements of the effectiveness of treatments between BSA (control) and PKA transfection (Figure 7 and Figure 8).

#### 3.4.1. Analysis of Synergy Effect of CRISPRMAX-Based PKA Delivery and DOX in Breast Cancer Cells

The breast cancer cells (MCF-7, BT-549, and Hs-578T) were transfected with 500 ng of PKA and treated with DOX at 0, 0.1, 0.5, 1, and 1.5 μg/mL at the same time for 48 h. The results of WST-1 analysis showed that the cell viability was significantly increased in MCF-7, BT-549, and Hs-578T (statically significant comparing with BSA, *p* = 0.0294, *p* = 0.0002, *p* = 0.0425, respectively) shown in Figure 7a,c,e. The cell viability greatly increased with the treatment with DOX following 12 h of PKA transfection in MCF-7 and BT-549 cells (statically significant compared with BSA, *p* = 0.0067, *p* = 0.0043, respectively) shown in Figure 7b,d. These results demonstrated the treatment with CRISPRMAX-PKAs along with DOX (neither at 0 h nor at 12 h of CRISPRMAX-based PKA delivery) did not improve the sensitivity to DOX but had opposite effects that caused the growth of breast cancer cells compared with controls.

#### 3.4.2. Analysis of Synergy Effect of CRISPRMAX-PKAs Delivery and PTX in Breast Cancer Cells

The breast cancer cells (MCF-7, BT-549, and Hs-578T) were transfected with 500 ng of PKA and treated with PTX at 0, 0.05, 0.1, 0.2, and 0.5 μg/mL at the same time for 48 h. When treating breast cancer cells with PTX and CRISPRMAX-PKAs simultaneously, the results of cell viability and AUC showed that the PKA group was greatly higher compared with the BSA group in BT-549 and Hs-578T cells (statically significant compared with the BSA group, *p* = 0.0004, *p* = 0.0088, respectively); as for MCF-7 cells, there was no difference between PKA and the control in these two cell lines (Figure 8a,c,e). The results showed that guiding PKAs through CRISPRMAX along with PTX at the same time provoked an increase of cell growth in BT-549 and Hs-578T. Whereas, adding PTX following 12 h of transfection with CRISPRMAX-PKAs in breast cancer cells, the cell viability was all remarkably decreased in MCF-7, BT-549, and Hs-578T (statically significant compared with the BSA group, *p* = 0.0018, *p* = 0.0290, *p* = 0.0031, respectively) shown in Figure 8b,d,f, which indicates that PKAs must be guided into breast cancer cells to trigger the repression of EMT before adding PTX, resulting in an increased sensitivity to PTX.

#### 3.4.3. Analysis of the Synergy Effect of CRISPRMAX-PKA Delivery and PTX in Normal Breast Cells

The side effect of chemotherapeutics lies in the non-selective killing of normal cells. In this study, we previously demonstrated that transfection with CRISPRMAX-PKAs can initiate an inhibition process that represses EMT and CSCs in breast cancer cells, thereby further weakening chemoresistance in breast cancer cells. Does the same phenomenon also occur in normal breast cells? To investigate whether PKA transfection can also affect sensitivity to chemotherapy drugs in normal cells, we delivered CRISPRMAX-PKAs along with PTX or DOX in normal breast epithelial cells MCF-10A. After 24 h of CRISPRMAX-PKAs delivery, different concentrations of PTX and DOX were added into the cells for a 48-h incubation. WST-1 was utilized for the detection of cell viability. The results demonstrated that CRISPRMAX-PKAs delivery did not alter the effects of the 48-h treatment with PTX or DOX in MCF-10A cells (Figure 9), which further confirmed that CRISPRMAX-PKAs delivery is specific to increase the sensitivity to PTX in mesenchymal breast cancer cells rather than in normal breast cells.

## 4. Discussion

In the field of cancer treatment, there are already some good enzyme drugs used in chemotherapy, such as arginine deaminase [35] and asparaginase [36,37], which utilize PEGylation to increase the half-life of exogenously infused recombinant enzymes and reduce immunogenicity. Our research concept is to try to apply nanotechnology in order to directly deliver the recombinant active enzymes inhibiting the EMT process into breast cancer cells. To our knowledge, this type of research has been conducted for the first time.

First of all, our study extends the application of CRISPRMAX for the delivery of functional enzymes besides Cas9 as a direct enzyme transfer strategy. Lipofectamine CRISPRMAX reagent, a lipid nanomaterial, was originally developed as a non-viral transfection with Cas9 enzymes and gRNA into the cells in order to edit the target gene [38]. In this study, instead of delivery of Cas9, we extended the application of CRISPRMAX for delivering functional EMT-inhibiting enzyme, which is PKA, into breast cancer cells. Considering the molecular weight of the enzyme encapsulated by CRISPRMAX, a classical PKA holoenzyme consists of two regulatory subunits (RI: 43–47 kDa or RII: 49–55 kDa) and two catalytic subunits (C: 40 kDa); thus, the molecular weight of PKA is comparable to that of Cas9 (around 163 kDa). The charge and nanoparticle size of CRISPRMAX-PKAs also have no differences compared with CRISPRMAX-BSAs. Our results show that the lipid nanoparticles (CRISPRMAX) can deliver fluorescent-labeled BSAs, which provides indirect evidence to support the notion that CRISPRMAX is able to deliver PKAs. Our immunofluorescence analysis of PKA expression further supports the notion that PKA enzymes can be delivered into breast cancer cells through CRISPRMAX. The establishment of this method helps to quickly introduce potential therapeutic proteins or enzymes into tumor cells in order to evaluate their therapeutic effects (around 3 to 7 days) instead of transcription or translation of the constructed nucleotide fragments, leading to the removal of risk of genomic integration.

Second, our study suggests a strategy that, combining EMT-targeting and the conventional chemotherapy, can improve the therapeutic effects of breast cancer treatment but requires careful design of the dosing regimen. EMT is a key biological process for promoting tumor cells to dedifferentiate to new CSCs after external stimulation, especially after radiotherapy and chemotherapy [39]. Many regulatory factors are involved in this process, including non-coding RNAs, proteins, enzymes, etc. When it comes to enzymes, most of them are overexpressed in the EMT process, which should be knockdown, and therefore are not suitable for the enzyme delivery treatment strategy. Recently, the cAMP/PKA signaling pathway was revealed to play a role in MET induction in mesenchymal breast cancer cells, which elicits their differentiation toward epithelial phenotypes, resulting in the loss of tumor-initiating capability [9]. Thus, we selected PKA as the model enzyme for investigating the biopharmaceutical effects of CRISPRMAX-encapsulation of functional enzymes. A recent study reported that the EMT process is not required for the development of lung metastasis but contributes to chemoresistance [40]. Therefore, the clinically relevant effect that we focused on in this study is chemoresistance. We selected two common chemotherapy drugs (paclitaxel and doxorubicin) with different mechanisms of action for combination therapy with PKA delivery, respectively. Paclitaxel is one of several cytoskeletal drugs that target tubulin. Paclitaxel-treated cells have defects in mitotic spindle assembly, chromosome segregation, and cell division [41,42]. Doxorubicin interacts with DNA by intercalation and inhibition of macromolecular biosynthesis [43]. Our study found that there is no synergistic effect of PKA and DOX, but a good synergistic effect of PKA and PTX on the inhibition of chemoresistance has been confirmed. Interestingly, in order to have good synergy effect of PKA and PTX, the breast cancer cells must be transfected with PKA for 12 h before adding PTX. If PTX and PKA are carried out at the same time, it will not only fail to inhibit tumor cells but even promote their growth. This indicates a beforehand process related to EMT inhibition or differentiation is necessary for such synergy (an overview diagram is shown in Figure 10).

When we use nanoparticles for combining EMT-targeting and conventional chemotherapy, we cannot always carry out the commonly used strategy that is the so-called “one-stop solution”, which loads two drugs in one nanoparticle at the same time [26], but need to consider taking a sequential solution strategy in terms of delivering combinational therapeutics in a single nanoparticle. Therefore, using nanoparticles for combinational therapy remains challenging. The significance of this study is that we discovered that delivering PKA enzymes first before administrating PTX can have a good synergy in EMT inhibition as well as chemosensitivity for breast cancer cells.

Why is there no synergy effect of PKA delivery and DOX, whereas adding PKA along with PTX has a good synergy? So far, the exact mechanism of such a phenomenon is not clear. Considering our results, PKA delivery inhibits EMT in breast cancer cells, resulting in a decrease in α-SMA expression. α-SMA is an actin protein that belongs to the actin cytoskeletons, and paclitaxel targets another type of cytoskeleton tubulin; based on these two facts, we hypothesized that the synergy effects of PKA and PTX may be related to cytoskeletal remodeling [44], which is also involved in driving the EMT process [45]. However, such a hypothesis still requires further investigation in the future.

Except for liposomes, there are also many other functionalized nanosystems that can be used to deliver enzymes, such as polymer nanoparticles [23], polymersomes [24], mesoporous nanoparticles [25], metal-organic framework (MOF) with biomineralization [26], DNA nanoparticles [27,28], emulsions [29], etc. In detail, for mesoporous silica nanoparticles (MSNs), it is reported that enzymes can be loaded in the cavity of the pores in a size-selective adsorption manner [46]. Polymers, including microspheres, polymer-coated substrates as small capsules (known as microencapsulation), hydrogels, and nanoparticles, can protect the protein drugs from premature degradation. Among these materials, polylactic acid (PLA) and poly(lactic-co-glycolic acid) (PLGA) are the most common biodegradable materials used in the development of protein microspheres [47]. Most importantly, a new kind of MOF, such as a zeolitic imidazolate framework, ZIF-8, can store biologically active enzymes in vitro for a long time [48]. In principle, enzyme-ZIF8 nanocomposition can be delivered into cells by endocytosis and release the delivered enzymes in a pH-dependent manner in the endosome/lysosome [49].

For characterization of PKAs- and BSAs-loaded CRISPRMAX complexes, dynamic light scattering (DLS) measurements were performed for analyzing the particle size, polydispersity index (PDI), and zeta potential of the enzymes-loaded CRISPRMAX complexes. In in vitro studies of using LNPs in therapeutics delivery, lipid-based particles with a PDI value of 0.3 or below are considered to be acceptable carriers for delivering therapeutics, as such vesicles are homogeneously distributed in the solution [50,51]; the PDI values of our enzymes-loaded CRISPRMAX complexes and null CRISPRMAX were also within this range, which indicated these complexes were evenly distributed in the solution. In addition, the average particle size of CRISPRMAX-PKAs, CRISPRMAX-BSAs, and null CRISPRMAX was approximately 350 nm; these complexes can be easily internalized by targeted cells, since their size is less than 500 nm [52]. Zeta potential can provide general information of the surface charge properties of nanoparticles; however, such an indicator has its limitations as the surface charge of nanoparticles can be significantly affected by the surrounding environment; small changes in any of these parameters, such as temperature, pH, conductivity (a parameter that determines the ionic strength of a solution), and viscosity of solvent, etc., have a profound impact on zeta potential value [53]. Therefore, our focus is not on the zeta potential value itself but to examine whether the zeta potential of the complexes is changed or not when loading different proteins through CRISPRMAX under the same condition. Our results showed that there was no significant difference in the surface charge of CRISPRMAX-PKAs and CRISPRMAX-BSAs complexes compared with control null CRISPRMAX. This is consistent with the results of Marija Brgles et al. that negatively charged protein could not influence the overall charge of liposomes [32]. Moreover, PKAs- and BSAs-loaded CRISPRMAX complexes can also be directly delivered to cells, as the lipid-based complexes can be delivered through cell internalization, directly fusing with the cell membrane [54]. This may be the reason why CRISPRMAX reagent has high transfection efficiency in cells.

Indeed, we have to admit the biggest shortcoming of such LNPs is that they cannot stably store target enzymes in vitro for a long time [12]. So, if considering potential administration in the clinic in the future, CRISPRMAX or CRISPRMAX-like reagent and enzymes (such as PKAs) should be stored separately at 4 and −20 °C, respectively. When administering the complex solution, PKA enzymes and CRISPRMAX can be mixed at room temperature before delivery into the targeted cancer cells. Solid nanoparticles, such as MOF, can store biologically active enzymes in vitro for a long time [26,48], of which the synthesis process is relatively easy to manipulate. However, compared with LNPs, it is not yet easy to perform a high-throughput preparation of enzyme-MOF complexes and functional screening in seeded cells. In short, it is necessary to choose these enzyme-encapsulation delivery strategies with each specific advantage according to the research purpose.

In summary, as a proof-of-concept, we confirmed the direct enzyme delivery of PKA as a potential strategy for inhibiting EMT/CSC-associated traits, including downregulation of the expression of EMT-related markers α-SMA and N-cad, chemoresistance, and mamospheres. PKA delivery has a significant inhibitory effect to PTX resistance but has no effect against DOX. The inhibitory effect of chemoresistance can be exerted only in the cells that were treated with PKA before PTX administration, which will shed light on the construction of a new drug delivery system or complex nanoparticles with a combinational therapy that targets both EMT/CSCs and bulk cancer cells. This direct enzyme delivery strategy will also facilitate the testing of target enzymes/proteins on their biological functions.

## Figures and Tables

**Figure 1 pharmaceutics-13-00011-f001:**
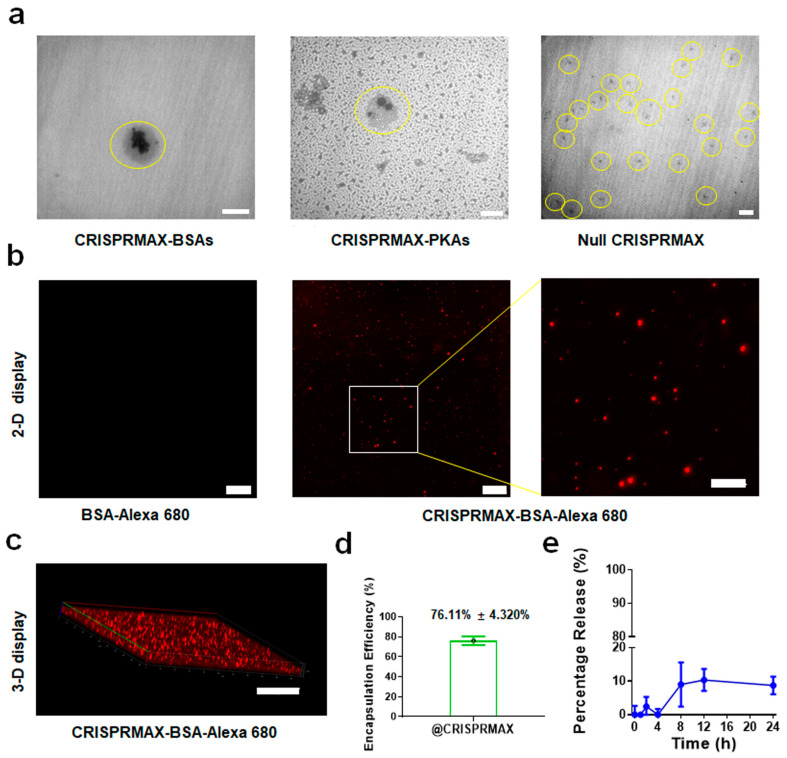
Characterization of PKAs- and BSAs-loaded CRISPRMAX complexes. (**a**) TEM images of CRISPRMAX-BSAs, CRISPRMAX-PKAs and null CRISPRMAX; the yellow circles indicate the location of the complexes (scale bars, 500 nm in BSAs and PKAs, and 1 μm in null complexes). (**b**) CLSM images of BSA-Alexa 680 and CRISPRMAX-BSA-Alexa 680 in a 2-D display under the same parameters; the right panel is a partial enlargement of CRISPRMAX-BSA-Alexa 680 in a 2-D display (scale bars, 20 μm in the left and middle panels; 10 μm in the right panel). (**c**) A CLSM image of CRISPRMAX-BSA-Alexa 680 in a 3-D display (scale bar, 50 μm). (**d**) The encapsulation efficiency of CRISPRMAX based on the calculation of loaded BSA-Alexa 680. (**e**) The release efficiency of CRISPRMAX based on the calculation of loaded BSA-Alexa 680 at the indicated time points.

**Figure 2 pharmaceutics-13-00011-f002:**
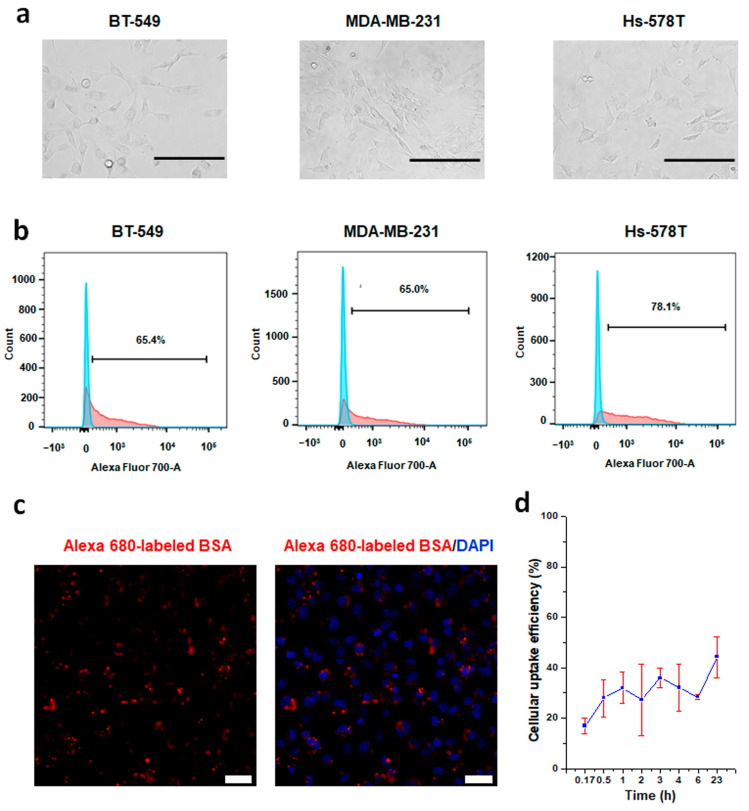
Establishment of a CRISPRMAX-based protein delivery system using the Alexa 680-labeled BSA as a model protein. (**a**) Bright-field images of three different mesenchymal breast cancer cell lines were taken at 200× magnification (scale bars, 200 μm). (**b**) Single-parameter histograms for comparison of the negative control (cells transfected with ddH_2_O, blue) and the positive population (the cell of interest, red): (**left**) BT-549 cells were seeded at the density of 8000 cells/well, and FCM was conducted at day 4 after transfection with fluorescent BSAs; (**middle**) MDA-MB-231 cells were seeded at the density of 4000 cells/well, and FCM was conducted at day 2 after transfection with fluorescent BSAs; (**right**) Hs-578T cells were seeded at the density of 8000 cells/well, and FCM was conducted at day 2 after transfection with fluorescent BSAs. Each kind of cell was transfected with 500 ng of BSA-Alexa Fluor 680 per well through CRISPRMAX. (**c**) CLSM analysis of BT-549 cells transfected with fluorescent BSAs delivered by CRISPRMAX. Fluorescent excitation and emission for BSA-Alexa Flour 680 were 633 and 702 nm, respectively. Confocal images were taken at 200× magnification (scale bars, 50 μm). (**d**) Cellular uptake efficiency of BSA-Alexa Fluor 680 delivered by CRISPRMAX within 24 h in BT-549 cells. CLSM, confocal laser scanning microscopy.

**Figure 3 pharmaceutics-13-00011-f003:**
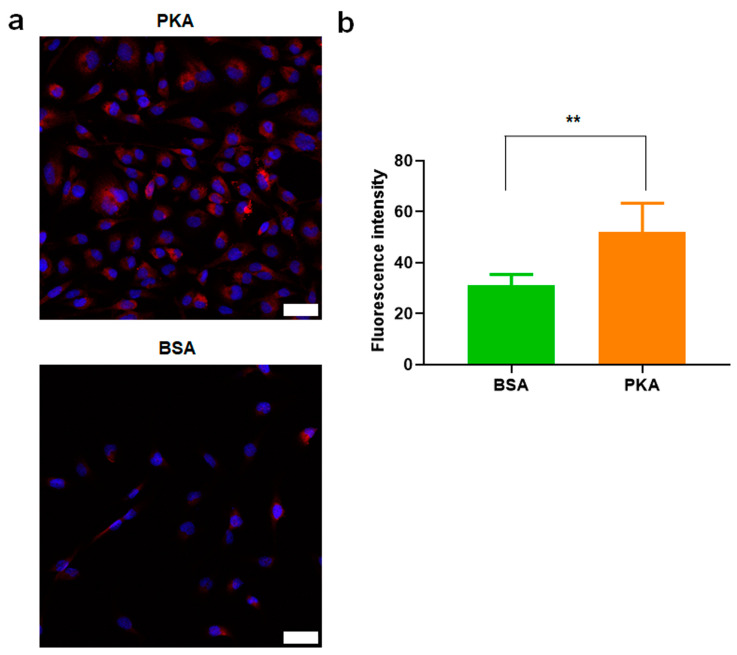
Immunofluorescence analysis of PKA expression in BT-549 cells after CRISPRMAX-based PKA delivery. (**a**) PKA expression was examined by immunofluorescence analysis using CLSM; 3 days after CRISPRMAX-based PKA delivery vs. CRISPRMAX-based BSA delivery (control) in BT-549 cells. Cell nuclei were visualized by DAPI. Confocal images were taken at 200× magnification (scale bars, 50 μm). (**b**) Quantitative analysis of the mean of florescence intensity of PKA performed by ImageJ (*n* = 3), and the mean of fluorescence intensity of PKA was significantly higher compared with BSA group, *p* = 0.00459, ** *p* < 0.01. The quantitative data were presented as Mean ± SD.

**Figure 4 pharmaceutics-13-00011-f004:**
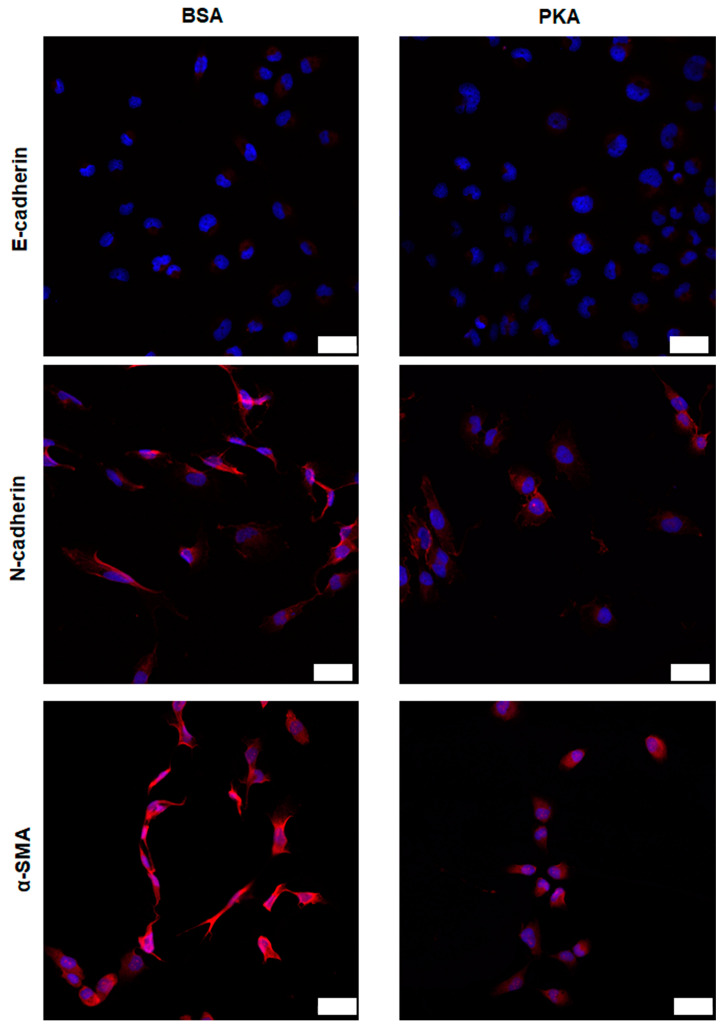
Expression analysis of E-cadherin, N-cadherin, and α-SMA after PKA delivery observed by CLSM. E-cadherin, N-cadherin, and α-SMA expression were examined by immunofluorescence analysis using confocal laser scanning microscopy 5 days after CRISPRMAX-based delivery, with the control (BSA) (left panel) vs. with PKA (right panel). Cell nuclei were visualized by DAPI. Confocal images were taken at 200× magnification (scale bars, 50 μm).

**Figure 5 pharmaceutics-13-00011-f005:**
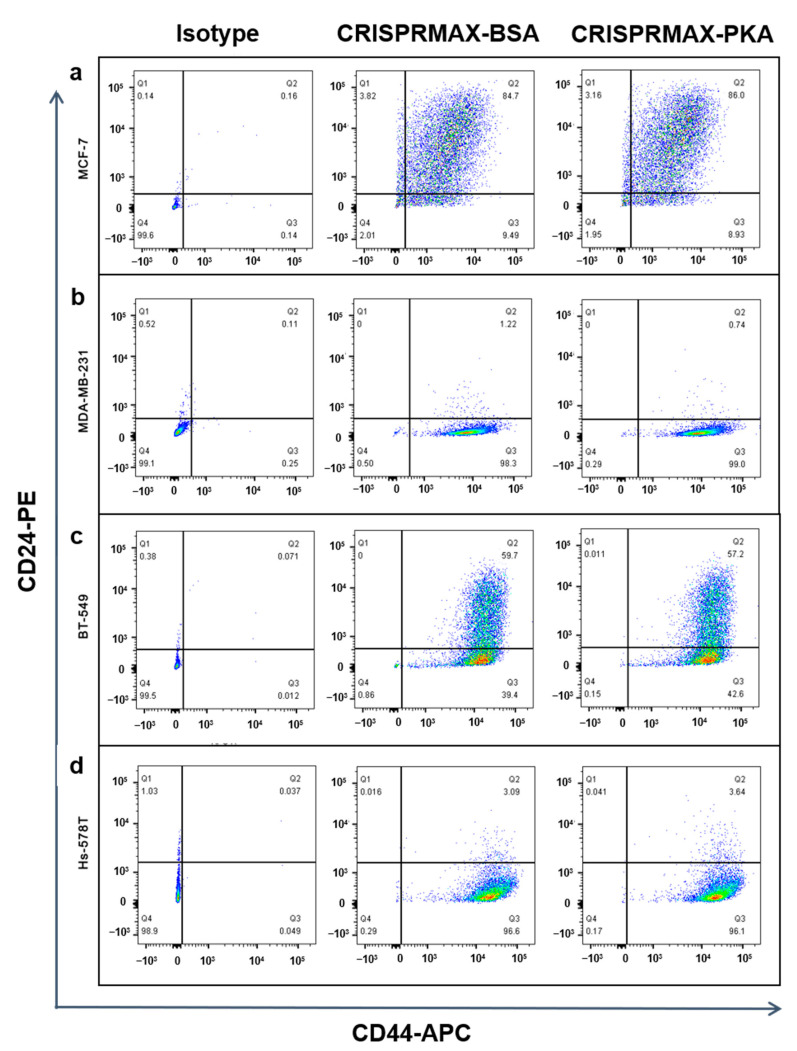
FCM analysis of the CD44^+^/CD24^−^ population in breast cancer cells after introducing CRISPRMAX-PKAs or CRISPRMAX-BSAs. (**a**) MCF-7 cells; (**b**) MDA-MB-231 cells; (**c**) BT-549 cells; (**d**) Hs-578T cells. The CD44^+^/CD24^−^ antigenic phenotype was shown in the Q3 area (at the right lower corner of the individual image). For isotype group: cells transfected with ddH_2_O through CRISPRMAX as controls were incubated with the same amount of APC-isotype IgG and PE-isotype IgG with CD44-APC and CD24-PE antibodies; as for the BSA and PKA groups, the cells transfected with BSAs or PKAs were incubated with CD44-APC and CD24-PE antibodies. *X*-axis presents CD44-APC of the cells; *Y*-axis shows CD24-PE of the cells. All cells were delivered with PKAs or BSAs for 3 to 5 days. FCM, flow cytometry.

**Figure 6 pharmaceutics-13-00011-f006:**
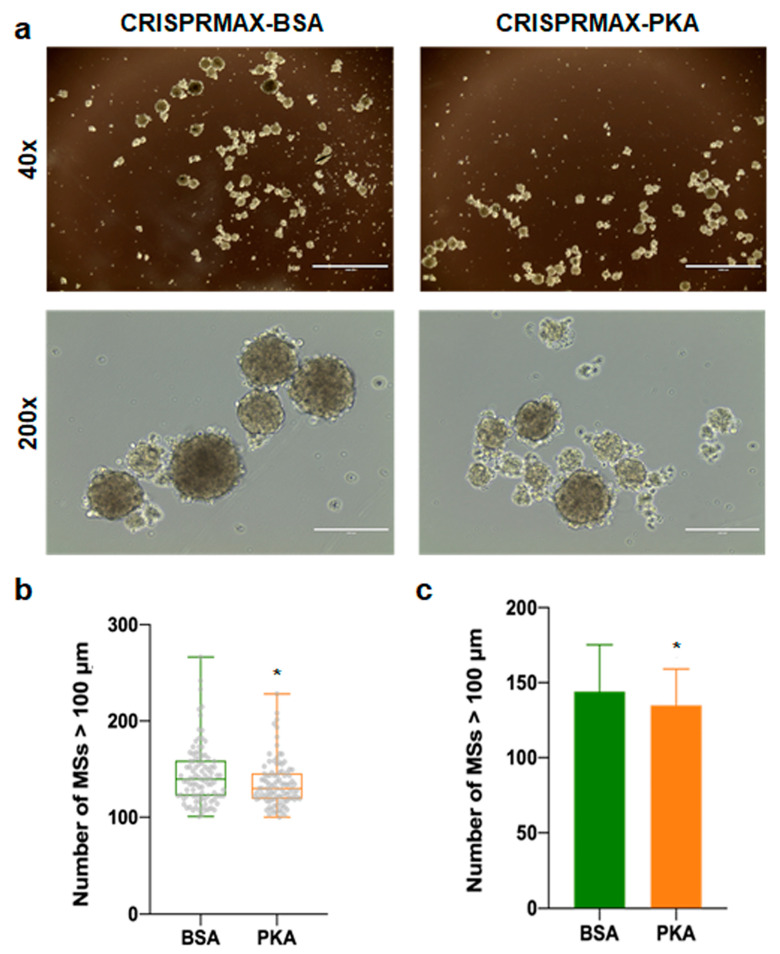
Mammosphere formation analysis of cells after CRISPRMAX-PKAs or CRISPRMAX-BSAs delivery. (**a**) Typical bright-field images of MSs were taken for each group (CRISPRMAX-PKAs and CRISPRMAX-BSAs) 1 week after culturing in mamosphere culture medium; for 40× and 200× magnification, the scale bar was 1000 and 200 μm, respectively. (**b**) Whisker-box plot of the number of MSs (diameter > 100 μm) for the CRISPRMAX-BSAs-delivered group and CRISPRMAX-PKAs-delivered group. (**c**) The bar graph of the number of MSs (diameter > 100 μm) for the CRISPRMAX-BSAs-delivered group and CRISPRMAX-PKAs-delivered group. The number of MSs in the PKA group was significantly lower compared with the BSA group, *p* = 0.0214, * *p* < 0.05. Data were presented as Mean ± SD.

**Figure 7 pharmaceutics-13-00011-f007:**
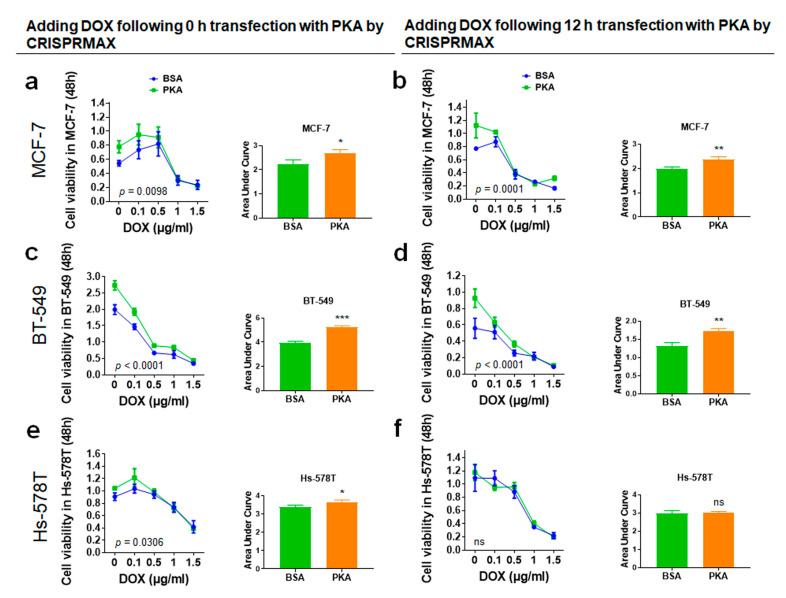
Chemoresistance evaluation of DOX in the cells following 0 or 12 h transfection with CRISPRMAX-PKAs. (**a**,**c**,**e**) Adding DOX following 0 h of transfection with CRISPRMAX-PKAs: MCF-7, BT-549, and Hs-578T cells were transfected with PKA by CRISPRMAX and treated with DOX at 0, 0.1, 0.5, 1, and 1.5 μg/mL simultaneously for 48 h; the results of AUC analyses in MCF-7, BT-549, and Hs-578T were compared with the BSA control group (*p* = 0.02094, *p* = 0.0002, *p* = 0.0425, respectively). (**b**,**d**,**f**) Cells were treated with DOX for 48 h following 12 h of transfection with CRISPRMAX-PKAs; the results of AUC analyses in MCF-7, BT-549, and Hs-578T were compared with the BSA control group (*p* = 0.0067, *p* = 0.0043, *p* = 0.8200, respectively). AUC analyses were used to compare the effectiveness of the drug effect between the BSA and PKA group. All treatments were performed with WST-1 cell viability assays and AUC analyses. Data were presented as Mean ± SD (*n* = 3 replicates for transfection with the PKA and BSA group). In total, 500 ng of PKA and BSA (control) per setup was used, and the PKA group was compared with the BSA control in each setup. Statistical significances were evaluated using unpaired two-tailed *t* test; * *p* < 0.05, ** *p* < 0.01, *** *p* < 0.001, and ns, not significant. DOX, doxorubicin.

**Figure 8 pharmaceutics-13-00011-f008:**
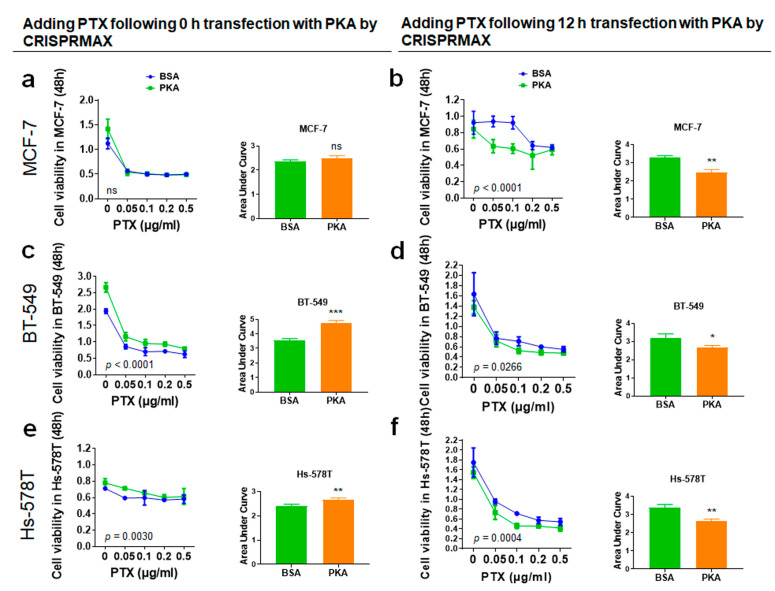
Chemoresistance assay of PTX in the cells following 0 or 12 h of transfection with CRISPRMAX-PKAs. (**a**,**c**,**e**) Adding PTX following 0 h of transfection with CRISPRMAX-PKAs: MCF-7, BT-549, and Hs-578T cells were transfected with PKA by CRISPRMAX and treated with PTX at 0, 0.05, 0.1, 0.2, and 0.5 μg/mL simultaneously for 48 h; the results of AUC analyses in MCF-7, BT-549, and Hs-578T were compared with the BSA control group (*p* = 0.1652, *p* = 0.0004, *p* = 0.0088, respectively). (**b**,**d**,**f**) Cells were treated with PTX for 48 h following 12 h of transfection with CRISPRMAX-PKAs; the results of AUC analyses in MCF-7, BT-549, and Hs-578T were compared with the BSA control group (*p* = 0.0018, *p* = 0.0290, *p* = 0.0031, respectively). AUC analyses were used to compare the effectiveness of drug effect between the BSA and PKA groups. All treatments were performed with WST-1 cell viability assays and AUC analyses. Data were presented as Mean ± SD (*n* = 3 replicates for transfection with the PKA and BSA group). In total, 500 ng of PKA and BSA (control) per setup was used, and the PKA group was compared with the BSA control in each setup. Statistical significances were evaluated using unpaired two-tailed *t* test; * *p* < 0.05, ** *p* < 0.01, *** *p* < 0.001, and ns, not significant. PTX, paclitaxel.

**Figure 9 pharmaceutics-13-00011-f009:**
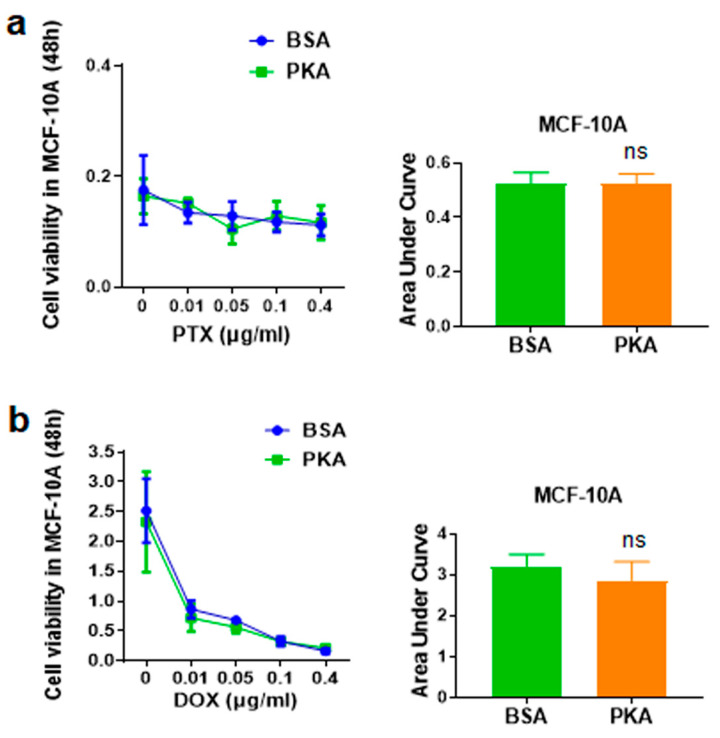
Cell viability results of normal breast cells MCF-10A treated with PTX or DOX following 24 h of transfection with CRISPRMAX-PKAs or CRISPRMAX-BSAs. (**a**) After 24 h of transfection with CRISPRMAX-PKAs, the MCF-10A cells were treated with PTX at 0, 0.01, 0.05, 0.1, and 0.4 μg/mL for 48 h. (**b**) After 24 h of transfection with CRISPRMAX-PKAs, MCF-10 cells were treated with DOX at 0, 0.01, 0.05, 0.1, and 0.4 μg/mL for 48 h; the results of AUC analyses in the PTX- or DOX-treated MCF-10A cells following 24 h of transfection with CRISPRMAX-PKAs were compared with the BSA control group (*p* = 0.9619, *p* = 0.3410, respectively). All treatments were performed with WST-1 cell viability assays and AUC analyses. AUC analyses were used to compare the effectiveness of the drug effect between the BSA and PKA groups. Data were presented as Mean ± SD (*n* = 3 replicates for transfection with the PKA and BSA group). In total, 500 ng of PKA and BSA (control) per setup was used, and the PKA group was compared with the BSA control in each setup. Statistical significances were evaluated using unpaired two-tailed *t* test; ns, not significant. PTX, paclitaxel; DOX, doxorubicin.

**Figure 10 pharmaceutics-13-00011-f010:**
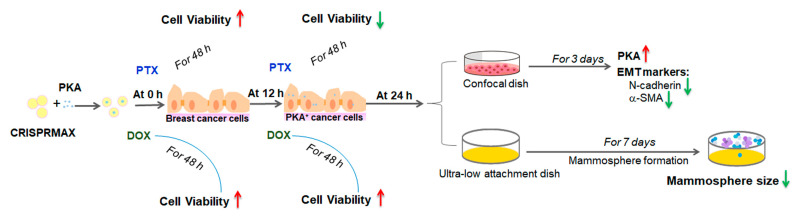
An overview schematic diagram of the study. PKAs and CRISPRMAX had formed as lipid nanoparticle complexes, and then these complexes were added to breast cancer cells. At the same time, chemotherapy drugs PTX or DOX were added (at 0 h transfection time point) into the cells for 48 h, respectively. The WST-1 assay was performed for testing the cell viability. The results showed that compared with the CRISPRMAX-BSAs control group added with the same drug, cell viability increased regardless of whether PTX or DOX was added. When CRISPRMAX-PKAs were added into cells for 12 h, the chemotherapy drug (PTX or DOX) was also added (at 12 h), respectively; the cells were treated for 48 h. WST-1 cell viability results showed that compared with the CRISPRMAX-BSAs control group added with the same drug, the cell viability of the PTX group was significantly decreased compared to the DOX group. Then, 24 h after CRISPRMAX-PKAs delivery into the cells, these cells were digested with trypsin and seeded in a confocal dish for 3 days, followed by immunofluorescence analysis of PKA expression. The results showed that the expression of PKAs was upregulated compared with the CRISPRMAX-BSAs control group; the expression of EMT mesenchymal markers, N-cadherin and α-SMA, was downregulated. Part of the cells were seeded in an ultra-low attachment dish at low density and subjected to mammosphere formation assay; these cells were cultured in mammosphere culture medium and continued to grow for 7 days, the results showed that CRISPRMAX-PKAs-delivered cells-derived mammospheres were decreased in size compared with the CRISPRMAX-BSAs control group.

**Table 1 pharmaceutics-13-00011-t001:** Size, polydispersity index (PDI), and zeta potential of optimized CRISPRMAX-PKAs, CRISPRMAX-BSAs, and control null CRISPRMAX. There was no significant difference in particle size compared with control null NPs (*p* = 0.88895, *p* = 0.38939, respectively). Zeta potentials were all around 8 mV negative charge, with no significant difference in the surface charge compared with control null NPs (*p* = 0.12596, *p* = 0.83723, respectively). Values were expressed as mean ± SD, *n* = 3.

Formulation	Size (nm)	PDI	Zeta Potential (mV)
CRISPRMAX-PKAs	349.8 ± 21.65	0.232 ± 0.172	−8.38 ± 0.337
CRISPRMAX-BSAs	344.5 ± 45.54	0.343 ± 0.051	−7.87 ± 0.0777
Null CRISPRMAX	345.56 ± 33.30	0.382 ± 0.231	−8.33 ± 0.403

## Data Availability

Data are contained within the article.

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
