# Peer review of "Delivery of Protein Kinase A by CRISPRMAX and Its Effects on Breast Cancer Stem-Like Properties"

_pharmaceutics, 2020, doi:10.3390/pharmaceutics13010011_

Round 1

Reviewer 1 Report

The article is well-written and structured.  It is interesting and good results have been achieved. Title and abstract reflect the content, while results are carefully explained and the conclusion is well supported by the data. I accept the manuscript with minor revision, such as:

Line 410, change “to directly” to “in order to directly”;

Line 487, use italics for the expression "in vitro";

Explain the method by which the LNPs were prepared;

Please correct the formatting and numbering of the manuscript. Moreover, it is necessary to center the images.

Author Response

Reviewer 1:

Comments: The article is well-written and structured.  It is interesting and good results have been achieved. Title and abstract reflect the content, while results are carefully explained and the conclusion is well supported by the data. I accept the manuscript with minor revision, such as:

Response:

Thanks for reviewer’s positive comments.

Comments: Line 410, change “to directly” to “in order to directly”;

Response:

Thanks for reviewer’s suggestions. We agree with that and have revised it. Please see it in Line 558, Page 24 of current revised version with “Track Changes” mode.

Comments: Line 487, use italics for the expression "in vitro";

Response:

Thanks for reviewer’s suggestions. We agree with that and have revised it. Please see it in Line 668, Page 26 of current revised version with “Track Changes” mode.

Comments: Explain the method by which the LNPs were prepared;

Response:

We are sorry for not clearly explaining the method by which the CRISPMAX-PKA complexes were prepared. In this work, we have utilized a commercial ready-to use product: CRISPRMAX (Lipofectamine™ CRISPRMAX™ Cas9 Transfection reagent, ThermoFisher Scientific), which is the first optimized lipid nanoparticle transfection reagent for CRISPR-Cas9 protein delivery (https://www.thermofisher.com/order/catalog/product/CMAX00008#/CMAX00008). To make it the step of CRISPMAX-PKA complexes preparation clearer, we have supplemented the detailed information into the section of method “BSA and PKA enzyme delivery by Lipofectamine CRISPRMAX Reagent” as below:

The previous description:

BSA and PKA enzyme delivery by Lipofectamine CRISPRMAX Reagent

Lipofectamine CRISPRMAX Reagent was used according to the manufacturer's protocol (Invitrogen). Briefly, Breast cancer cells and MCF-10A cells were placed in a 96-well plate (5, 000 cells per well, except for MDA-MB-231 cells with 2, 000 cells per well) one day before transfection. When delivering, prepare tube 1 containing Opti-MEMTM medium and PKA enzyme solution (500ng per well, 1μg/μl stock solution) with the same amount of purified BSA or Alex Fluor 680-labeled BSA (Invitrogen) as the controls. Tube 2 contained Opti-MEMTM medium with CRISPRMAX reagent. Next, immediately add solution from Tube 1 to Tube 2 within 3 min, then mix well, followed by incubating the complex solution at room temperature. After 5-10 min, 10 μL/well of the LNPs complex solution is dropped into the cancer cells medium (100 μL/well) for incubating for one day. Then the LNPs-contained medium was replaced and continued growing 1-3 days before visualizing/analyzing the transfected cells.”

The revised description:

Enzyme delivery by Lipofectamine CRISPRMAX Transfection Reagent

Lipofectamine CRISPRMAX reagent (Invitrogen) was used for delivering BSA and PKA enzymes in this study. All procedures were followed according to the manufacturer's protocol. Breast cancer cells and MCF-10A cells were seeded in 96-well plates (5000 cells per well, except for MDA-MB-231 cells were seeded at the density of 2000 cells per well) one day before transfection. The seeded plates were incubated overnight; the cells were grown at 50-70% confluence in 96-well-plate for transfection. On the day of transfection, the culture medium of each well in the plates was replaced with 100 μL of Opti-MEM before performing CRISPRMAX-based enzyme delivery. As for preparation of enzymes-loaded CRISPRMAX, Tube 1 contained 5 μL Opti-MEMTM medium (Invitrogen) and add 0.5 μL PKA enzyme solution (500ng per well, 1μg/μl stock solution) and 0.5 μL Cas9 Plus Reagent in sequence. The control cells were used the same amount of purified BSA, or Alex Fluor 680-labeled BSA (Invitrogen). Tube 2 contained 5 μL Opti-MEMTM medium with 0.5 μL CRISPRMAX reagent. Next, solution of Tube 1 was added to Tube 2 within 3 minutes, and then mixed well; the complex solution was incubated at room temperature. After 5-10 min, 10 μL/well of the CRISPRMAX complex solution was added into the cancer cells for 1 day incubation. Then, the CRISPRMAX-contained medium was replaced with complete growth medium in each well; the transfected cells continued to grow for 1 to 3 days before conducting further experiments.”

We have marked the changes using the track changes mode in the new revised manuscript, please see it in  Line 163-186, Page 4.

Comments: Please correct the formatting and numbering of the manuscript. Moreover, it is necessary to center the images.

Response:

Thanks for reviewer’s questions. We have corrected the formatting and numbering of the manuscript and made the images in the center of the page. Please see it in Line 149, 163, 188 of Page 4;  Line 219, Line 242 of Page 5; Line 257, Line 271, Line 292 of Page 6; Fig. 1 in Page 9; Fig. 2 in Page 11; Fig. 3 in Page 14; Fig. 4 in Page 17; Fig. 5 in Page 18; Fig. 6 in Page 20; Fig. 7 in Page 22; Fig. 8 in Page 23; Fig. 9 in Page 25.

Reviewer 2 Report

This is an interesting which could be valuable for the readership of pharmaceutics. I suggest it should be considered for publication once the following changes have been made and other points have been addressed satisfactorily

  1. Please briefly give the rationale in the abstract why PKA Enzyme is important in cancer therapy at the beginning of the section
  2. Why did the authors select CRISPMAX reagent over others?
  3. Please define all terms (including transcription factors (early on) if possible
  4. Please elaborate further on the various types of particulate systems that can be used to deliver the enzyme
  5. Please indicate how such LNS could be stored and administered
  6. Finally, please check English style and prose as it flicks between present and past tense

Author Response

Reviewer2:

Comments: This is an interesting which could be valuable for the readership of pharmaceutics. I suggest it should be considered for publication once the following changes have been made and other points have been addressed satisfactorily

Response:

Thanks for reviewer’s positive comments.

Comments:  1. Please briefly give the rationale in the abstract why PKA Enzyme is important in cancer therapy at the beginning of the section

Response:

Thanks for reviewer’s question. It is really very important to clearly elucidate the role of PKA in cancer therapy. Because of the word count limit for abstract (about 200 words maximum according to journal guideline), I have revised the first sentence of abstract “Protein kinase A (PKA) is an important enzyme for regulating epithelial-to-mesenchymal transition (EMT)-related activities.” into “Protein kinase A (PKA) activation is recently reported to inhibit epithelial-mesenchymal transition (EMT) and cancer stem cell (CSC) ability, which is considered to be responsible for chemoresistance and tumor recurrence of patients.” in Line 28-32 in Page 1 of current version of manuscript with “Track changes” mode.

Comments: 2. Why did the authors select CRISPMAX reagent over others?

Response:

Thanks for reviewer’s question. The main focus for this study is the biological understanding of a direct delivery of PKA enzyme into the cancer cells. The PKA has the function of inhibiting epithelial-to-mesenchymal transition and cancer stem cell, thus it has great potential in cancer treatment including chemoresistance. In this work, we have utilized a commercial ready-to-use product: CRISPRMAX (Lipofectamine™ CRISPRMAX™ Cas9 Transfection Reagent, ThermoFisher SCIENTIFIC), which is the first optimized lipid nanoparticle transfection reagent for CRISPR-Cas9 protein delivery (https://www.thermofisher.com/order/catalog/product/CMAX00008#/CMAX00008) (Biotechnol Lett 2016, 38, 919-929; Nat Commun 2020, 11, 4043; Leukemia. 2020 May 12;10.1038/s41375-020-0856-3.), since this is a very well established method, and the results obtained by this commercial kit will provide a reference for broader biological community and it will be easily repeated by other labs, even they have no material science background. This will also be a practically handled method for clinical doctors, for future clinical applications.

In fact, there are many other more functionalized nanosystems that can be used to deliver protein/enzyme. For example, in our research group, we have developed many different systems for protein delivery, including: (a) Liposomes (Adv. Funct. Mater., 2015, 25, 3330-3340. doi: 10.1002/adfm.201500594), (b) Polymer nanoparticles (Adv. Funct. Mater., 2017, 27 (42), 1703303. doi: 10.1002/adfm.201703303), (c) Polymersomes (Proc Natl Acad Sci U S A. 2019 Apr 16;116(16):7744-7749. doi: 10.1073/pnas.1817251116.), (d) Mesoporous nanoparticles (Adv. Funct. Mater. 2019, 29(43), 1902652. doi: 10.1002/adfm.201902652), (e) Metal organic framework with biomineralization (Anal Chem. 2020 Aug 18;92(16):11453-11461. doi: 10.1021/acs.analchem.0c02599.), (f) DNA nanoparticles (Adv Healthc Mater. 2017 Sep;6(18). doi: 10.1002/adhm.201700692; Adv Mater. 2018 Jun;30(24):e1703658. doi: 10.1002/adma.201703658.), (f) emulsions (Adv Mater. 2016 Dec;28(46):10195-10203. doi: 10.1002/adma.201602763.), etc.

But in the end, we select the commercial one CRISPRMAX, because CRISPRMAX has the advantages as below: (a) Low toxicity to cells, allowing us to focus on evaluating the biological effects of PKA enzyme on cells, including epithelial-mesenchymal transition, cancer stem cell and chemoresistance; (b) The experimental conditions have been optimized. There is a standard protocol for CRISPRMAX-enzyme preparation and cell transfection, which is involved in cell seeding density, reagent usage, the time of each step and the precautions for each step; (c) High throughput friendly—an ideal delivery solution for 96-well format. These advantages make us easily focus on the biological function of a direct delivery of PKA enzyme into the cancer cells rather than nanocarriers themselves and provide a good paradigm of potential translational application in clinics. Therefore, we select Lipofectamine CRISPRMAX reagent as a model nanocarrier for investigating establishment of direct transfer of PKA into breast cancer cells in the study.

We have supplemented the corresponding content above into the section of introduction of the manuscript Line 653-665, Page 26.

Comments:  3. Please define all terms (including transcription factors (early on) if possible

Response:

Thanks for reviewer’s question. We have defined all terms as the reviewer suggested, including: transcription factors including Snail Family Transcriptional Repressor 1 (SNAIL1), Twist Family BHLH Transcription Factor 1 (TWIST), Zinc Finger E-Box Binding Homeobox 1 (ZEB1), microRNAs (miRNAs), CRISPRMAX (Lipofectamine™ CRISPRMAX™ Cas9 Transfection Reagent), bovine serum albumin (BSA), flow cytometry (FCM), α-smooth muscle actin (α-SMA), N-cadherin (N-cad), mammospheres (MSs), paclitaxel (PTX), and doxorubicin (DOX) in “Introduction” section, and confocal laser scanning microscopy (CLSM) in “Results” section.

Comments: 4. Please elaborate further on the various types of particulate systems that can be used to deliver the enzyme

Response:

Thanks for reviewer’s question. We agree with the reviewer that there are many other more functionalized nanosystems that can be used to deliver the enzyme, such as liposomes, polymers, mesoporous nanoparticles, etc. which have great potential as a carrier for the protein delivery. In our research group, we have also developed many different systems for protein delivery, including: (a) Liposomes (Adv. Funct. Mater., 2015, 25, 3330-3340. doi: 10.1002/adfm.201500594), (b) Polymer nanoparticles (Adv. Funct. Mater., 2017, 27 (42), 1703303. doi: 10.1002/adfm.201703303), (c) Polymersomes (Proc Natl Acad Sci U S A. 2019 Apr 16;116(16):7744-7749. doi: 10.1073/pnas.1817251116.), (d) Mesoporous nanoparticles (Adv. Funct. Mater. 2019, 29(43), 1902652. doi: 10.1002/adfm.201902652), (e) Metal organic framework with biomineralization (Anal Chem. 2020 Aug 18;92(16):11453-11461. doi: 10.1021/acs.analchem.0c02599.), (f) DNA nanoparticles (Adv Healthc Mater. 2017 Sep;6(18). doi: 10.1002/adhm.201700692; Adv Mater. 2018 Jun;30(24):e1703658. doi: 10.1002/adma.201703658.), (f) emulsions (Adv Mater. 2016 Dec;28(46):10195-10203. doi: 10.1002/adma.201602763.), etc.

In detail, the CRISPRMAX we used in the study belongs to a kind of liposomes. For mesoporous silica nanoparticles (MSN), it is reported that enzymes can be loaded in the cavity of the pores in size-selective adsorption manner (Smart Mesoporous Silica Nanoparticles for Protein Delivery. Liu HJ, et al. Nanomaterials (Basel). 2019). For polymers including microsphere, microcapsulation, hydrogels, nanoparticles, they can protect the protein drugs from premature degradation. Among these materials, poly lactic acid (PLA) and poly lactic-co-glycolic acid (PLGA) are the most common biodegradable materials used in the development of protein microspheres (Recent Advance in Polymer Based Microspheric Systems for Controlled Protein and Peptide Delivery. Current Medicinal Chemistry, 2019, 26, 2285-2296). Most important, a new kind of metal organic framework (MOF), such as a zeolitic imidazolate framework, ZIF-8, can store biologically active enzymes in vitro for a long time (Biomimetic mineralization of metal-organic frameworks as protective coatings for biomacromolecules. Nat Commun. 2015 Jun 4;6:7240.). In principle, enzyme-ZIF8 nanocomposition can be delivered into cells by endocytosis and released the delivery enzyme in a pH-dependent manner in endosome/lysosome (Packaging and delivering enzymes by amorphous metal-organic frameworks. Nat Commun. 2019 Nov 14;10(1):5165.). In summary, it is necessary to choose these enzyme-encapsulation deliver strategies with each specific advantage according to the research purpose.

We have also supplemented the content above into the section of discussion of the manuscript in Line 653-665, Page 26.

Comments: 5. Please indicate how such LNS could be stored and administered

Response:

Thanks for reviewer’s question. The storing is really very important for future application. Like other liposome reagents, CRISPRMAX-enzyme cannot be stored in vitro for a long time as well as the company’s standard protocol suggested "Incubate complex for 5-10 minutes at room temperature. Do not incubate for >30 minutes". It may not be able to effectively deliver into cells after storage of the complex for more than 30 minutes. So, if considering potentially administered in clinics in future, the CRISPRMAX or CRISPRMAX-like reagent and enzyme (such as PKA) should be stored separately. According to the manufacture’s datasheet, CRISPRMAX should be stored in 4 oC, do not freeze. When stored at -20 °C in the form of dry solid, the PKA enzyme will lose <10% activity per year. After the first use, the stock solution of PKA can be stored in aliquots at -20 oC. When administered, the PKA enzyme and CRISPRMAX can be mixed according to the manufacture’s protocol in room temperature before delivered into the target cancer cells.

We have also supplemented the related content above into the section of discussion of the manuscript in Line 672-676, Page 26.

Comments: 6. Finally, please check English style and prose as it flicks between present and past tense

Response:

Thanks for reviewer’s suggestions. We have extensively checked the manuscript thoroughly and corrected the typos and mistakes by ourselves and together with an English native speaker. The corrections are marked with “Track Changes” mode in the manuscript.

Reviewer 3 Report

The manuscript submitted by Guo, Eriksson, Zhang et al. reports the generation of PKA-loaded lipid nanoparticles and the assessment of the effectiveness of these novel nanodevices for the treatment of breast cancer. The strategy proposed relies on the control delivery of PKA enzyme by the use of nanotechnology for inhibiting EMT/CSC-associated traits including chemoresistance. As a proof of concept, the manuscript compiles the results obtained using very well-known therapeutic agents such as DOX and PTX.

The strategy is deemed to be far more original. As far as I know, there is anything related reported at this moment. However, the results are very preliminary, and it is quite difficult to draw conclusions. In my opinion, the authors have made an important effort to analyze cell uptake efficiency and the effects of the novel nanovedices but they have overlooked nanotechnology. They have not properly explained the nanoparticle formulation. The formulation is not optimized (the sizes of the nanoparticles are very high, the PDI have to be improved and the Z potential values indicate unstable formulations). There is anyting concerning to EE and LE values. The morphology of the nanodevices were not studied by TEM or SEM. Release studies in vitro are missed and also stability studies after storage.

Please, it will be also neccesary to revise the manuscript again before submiting elsewhere. There are some important typos and mistakes.

I am afraid, I cannot suggest this work to be published in pharmaceutics when all this important information is missing. It is difficult to agree with the conclusions drawn by the authors.     

Author Response

Reviewer 3

The manuscript submitted by Guo, Eriksson, Zhang et al. reports the generation of PKA-loaded lipid nanoparticles and the assessment of the effectiveness of these novel nanodevices for the treatment of breast cancer. The strategy proposed relies on the control delivery of PKA enzyme by the use of nanotechnology for inhibiting EMT/CSC-associated traits including chemoresistance. As a proof of concept, the manuscript compiles the results obtained using very well-known therapeutic agents such as DOX and PTX.

Comments: The strategy is deemed to be far more original. As far as I know, there is anything related reported at this moment.

Response:

Thanks for reviewer’s comments. Indeed, to our knowledge, our direct PKA enzyme delivery strategy for inhibiting epithelial-to-mesenchymal transition (EMT)/cancer stem cell (CSC)-associated traits has been conducted for the first time.

Comments: However, the results are very preliminary, and it is quite difficult to draw conclusions. In my opinion, the authors have made an important effort to analyze cell uptake efficiency and the effects of the novel nanovedices but they have overlooked nanotechnology. They have not properly explained the nanoparticle formulation. The formulation is not optimized (the sizes of the nanoparticles are very high, the PDI have to be improved and the Z potential values indicate unstable formulations). There is anyting concerning to EE and LE values. The morphology of the nanodevices were not studied by TEM or SEM. Release studies in vitro are missed and also stability studies after storage. 

Response:

We thank the reviewer for the comments. The main focus for this study is the biological understanding of a direct delivery of PKA enzyme into the cancer cells. The PKA has the function of inhibiting epithelial-to-mesenchymal transition and cancer stem cell, thus it has great potential in cancer treatment including chemoresistance. In this work, we have utilized a commercial ready-to use product: CRISPRMAX (Lipofectamine™ CRISPRMAX™ Cas9 Transfection Reagent, ThermoFisher SCIENTIFIC), which is the first optimized lipid nanoparticle transfection reagent for CRISPR-Cas9 protein delivery (https://www.thermofisher.com/order/catalog/product/CMAX00008#/CMAX00008), since this is a very well established method, and the results obtained by this commercial kit will provide a reference for broader biological community and it will be easily repeated by other labs, even they have no material science background. This will also be a practically handled method for clinical doctors, for future clinical applications.

For the nanoparticle formulation, we have followed the instruction from the commercial kit (Fig. 1), but we have also done the following improvement, as listed below:

  • Because we don’t need gRNA in our PKA enzyme-delivery strategy, we modified the standard protocol by deleting gRNA
  • We increased PKA enzyme usage up to 500ng per well, and it is working well after repeating for many times.
  • To make the step clearer, we have supplemented the detailed information into the section of method “BSA and PKA enzyme delivery by Lipofectamine CRISPRMAX Reagent” as below:

The previous description:

BSA and PKA enzyme delivery by Lipofectamine CRISPRMAX Reagent

Lipofectamine CRISPRMAX Reagent was used according to the manufacturer's protocol (Invitrogen). Briefly, Breast cancer cells and MCF-10A cells were placed in a 96-well plate (5, 000 cells per well, except for MDA-MB-231 cells with 2, 000 cells per well) one day before transfection. When delivering, prepare tube 1 containing Opti-MEMTM medium and PKA enzyme solution (500ng per well, 1μg/μl stock solution) with the same amount of purified BSA or Alex Fluor 680-labeled BSA (Invitrogen) as the controls. Tube 2 contained Opti-MEMTM medium with CRISPRMAX reagent. Next, immediately add solution from Tube 1 to Tube 2 within 3 min, then mix well, followed by incubating the complex solution at room temperature. After 5-10 min, 10 μL/well of the LNPs complex solution is dropped into the cancer cells medium (100 μL/well) for incubating for one day. Then the LNPs-contained medium was replaced and continued growing 1-3 days before visualizing/analyzing the transfected cells.”

The revised description:

Enzyme delivery by Lipofectamine CRISPRMAX Transfection Reagent

Lipofectamine CRISPRMAX reagent (Invitrogen) was used for delivering BSA and PKA enzymes in this study. All procedures were followed according to the manufacturer's protocol. Breast cancer cells and MCF-10A cells were seeded in 96-well plates (5000 cells per well, except for MDA-MB-231 cells were seeded at the density of 2000 cells per well) one day before transfection. The seeded plates were incubated overnight; the cells were grown at 50-70% confluence in 96-well-plate for transfection. On the day of transfection, the culture medium of each well in the plates was replaced with 100 μL of Opti-MEM before performing CRISPRMAX-based enzyme delivery. As for preparation of enzymes-loaded CRISPRMAX, Tube 1 contained 5 μL Opti-MEMTM medium (Invitrogen) and add 0.5 μL PKA enzyme solution (500ng per well, 1μg/μl stock solution) and 0.5 μL Cas9 Plus Reagent in sequence. The control cells were used the same amount of purified BSA, or Alex Fluor 680-labeled BSA (Invitrogen). Tube 2 contained 5 μL Opti-MEMTM medium with 0.5 μL CRISPRMAX reagent. Next, solution of Tube 1 was added to Tube 2 within 3 minutes, and then mixed well; the complex solution was incubated at room temperature. After 5-10 min, 10 μL/well of the CRISPRMAX complex solution was added into the cancer cells for 1 day incubation. Then, the CRISPRMAX-contained medium was replaced with complete growth medium in each well; the transfected cells continued to grow for 1 to 3 days before conducting further experiments.”

Round 2

Reviewer 3 Report

I am afraid, my opinion did not change after the improvements proposed by the authors. If the authors decided to use nanotechnology they should have worried about the optimization of the formulations before carrying out any biological assays. As I told before, the nanodevices are not optimized enough to be published in pharmaceutics.

Author Response

We thank the reivewer for his/her comments. We still don’t think the characterizations of CRISPRMAX reagent, including TEM, encapsulation efficiency and release properties, are necessary for this manuscript, since CRISPRMAX reagent is a well-established commercial Cas9 enzyme delivery reagent developed by Thermofisher Scientific, and those are product information that should be provided by Thermofisher Scientific; and we just use it as a commercial kit to do a biological study. This work has very little to do with nanoparticle development.

But since we have all facilities on conducting these experiments based on our lab’s nanotechnology platforms, we have done what the review has required and provided the data by a point-to-point manner. The methods and the results of the experiments mentioned above were also added into the corresponding context.

We include the newly collected experimental data in the new figure 1. The original figure 1 becomes the new figure 2, and so on. Now, there is in total 10 figures in the revised manuscript. The other unified format changes in the text and figures, such as font size in the figure, vertical and horizontal rows of text in the figures, serial number adjustment and modification, uniform horizontal coordinate format, uniform p value in the figure with *, etc., which are revised in the second round of revision, are all marked with “tracking changes” mode in the manuscript.

The following are point-to-point responses to reviewers’ comments

Reviewer Comments: The work presented in this manuscript is novel and is thus suited for publication in Pharmaceutics. However, the authors have to provide more rigorous characterization of their delivery system before this manuscript can be accepted for publication. In particular, the authors have not addressed the questions raised by Reviewer 3: "They have not properly explained the nanoparticle formulation. The formulation is not optimized (the sizes of the nanoparticles are very high, the PDI have to be improved and the Z potential values indicate unstable formulations). There is anyting concerning to EE and LE values. The morphology of the nanodevices were not studied by TEM or SEM. Release studies in vitro are missed and also stability studies after storage."The authors should provide a discussion around the DLS measurements of their complexes in comparison with pertinent literature, and provide data regarding the morphology. Also encapsulation efficiency and release properties should be addressed.  

Comment 1: “The authors should provide a discussion around the DLS measurements of their complexes in comparison with pertinent literature”

Response 1:

For characterization of PKAs- and BSAs-loaded CRISPRMAX complexes, dynamic light scattering (DLS) measurements were performed for analyzing the particle size, polydispersity index (PDI) and zeta potential of the enzymes-loaded CRISPRMAX complexes. In in vitro studies of using LNPs in therapeutics delivery, lipid-based particles with a PDI value of 0.3 or below are considered to be acceptable carriers for delivering therapeutics, as such vesicles are homogeneously distributed in the solution [51, 52]; the PDI values of our enzymes-loaded CRISPRMAX complexes and null CRISPRMAX were also within this range, which indicated these complexes were evenly distributed in the solution. In addition, the average particle size of CRISPRMAX-PKAs, CRISPRMAX-BSAs and null CRISPRMAX was approximately 350 nm; these complexes can be easily internalized by targeted cells, since their size is less than 500 nm [53]. Zeta potential can provide general information of surface charge properties of nanoparticles; however, such indicator has its limitations as the surface charge of nanoparticles can be significantly affected by the surrounding environment; small changes in any of these parameters, such as temperature, pH, conductivity (a parameter that determines the ionic strength of a solution), and viscosity of solvent, etc., have a profound impact on zeta potential value [54]. Therefore, our focus is not on the zeta potential value itself, but to examine whether the zeta potential of the complexes is changed or not when loading different proteins through CRISPRMAX under the same condition. Our results showed that there were no significant differences in the surface charge of CRISPRMAX-PKAs, CRISPRMAX-BSAs complexes compared with the control null CRISPRMAX. This is consistent with the results of Marija Brgles et al. that negatively charged protein could not influence the overall charge of liposomes [55]. Moreover, PKAs- and BSAs-loaded CRISPRMAX complexes can also be directly delivered to cells; as the lipid-based complexes can be delivered through cell internalization, directly fusing with the cell membrane [56]. This may be the reason why CRISPRMAX reagent has high transfection efficiency in cells.

Comment 2: “……and provide data regarding the morphology.”

Response 2:

The morphology data of the enzymes-loaded CRISPRMAX complexes were collected by means of transmission electron microscopy (TEM) and confocal laser scanning microscopy (CLSM) as follows: for TEM analysis, the fresh CRISPRMAX-BSA complexes, CRISPRMAX-PKA complexes, and null CRISPRMAX complexes were prepared, respectively, according to the protocol mentioned in Section 2.2. The grids used for the electron microscope samples were immersed in the complex solution for each setup for a few seconds, then taken out, and placed on an absorbent paper for drying overnight. Under TEM, the different appearances of CRISPRMAX and the encapsulated proteins were clearly shown. CRISPRMAX was shown as a light gray outer outline of the complexes, while the encapsulated BSA or PKA proteins were presented in black (shown in Figure 1a as below).

For null CRISPRMAX, there was only a light gray outline. These results showed that the proteins were successfully coated by CRISPRMAX. Meanwhile, for CLSM analysis, fluorescent BSAs (Alexa Fluor 680-labeled BSAs) were used to prepare fresh CRISPRMAX-BSA-Alexa 680 complexes; the same concentration of BSA-Alexa 680 solution without CRISPRMAX was used as a control group. The solution of the complexes or the control group was dripped onto the glass slides, and then these samples were covered with cover glasses. Fluorescent particles were clearly observed under CLSM; as the images in 2D and 3D display showed that small red particles were distributed in the solution, while BSA-Alexa 680 in the control group had no significant red particles due to lack of CRISPRMAX encapsulation (Figure 1b,c), which further indicates BSA-Alexa 680 proteins were successfully coated by CRISPRMAX.

Comment 3: “Also encapsulation efficiency should be addressed.

Response 3:

Fluorescent-labeled BSAs (BSA-Alexa 680) were used as model to mimetic the encapsulation efficiency of CRISPRMAX, since it is very difficult to directly quantify PKA. Briefly, the fresh CRISPRMAX-BSA-Alexa 680 complexes were prepared, according to the protocol mentioned in Section 2.2. In the meantime, the same amount of BSA-Alexa 680 without CRISPRMAX was prepared in the same volume of solution as positive control groups, of which the fluorescence intensity represented the total amount of BSA-Alexa 680 (Vt). Then, the complex solution was added into the upper chamber of the centrifugal filter device (100K, Merck Millipore), followed by centrifugation at 13000 rpm for 5 min. The molecular weight of CRISPRMAX and CRISPRMAX-BSA-Alexa 680 complexes is greater than 100 kDa, andthe complexes will be trapped in the filter, while the molecular weight of unencapsulated free BSA-Alexa 680 is less than 100 kDa, and will be passed through the filter. After centrifugation, the filtrate was collected. Through this method, the unencapsulated free BSAs in the preparation system were obtained. The fluorescence intensity of these complexes was analyzed with a Varioskan multimode reader, and the fluorescence intensity of BSA-Alexa 680 (Vx) was obtained. The encapsulation efficiency was calculated based on the following formula:

Encapsulation efficiency (%) = (Vt − Vx) / Vt × 100%

The results showed that our encapsulation efficiency using CRISPRMAX to encapsulate the protein was 76.11% ± 4.320% (Figure 1d).

Comment 4: “……release properties should be addressed.

Response 4:

For in vitro release study of CRISPRMAX, fluorescent BSAs (BSA-Alexa 680) were used to calculate the release efficiency of CRISPRMAX. The release properties were measured at the following time points: 0 h, 1 h, 2 h, 4 h, 8 h, 12 h, and 24 h. First, the fresh CRISPRMAX-BSA-Alexa 680 complexes were prepared, according to the protocol mentioned in Section 2.2; 1.25 μg of BSA-Alexa 680 in 250 μL of complex solution per tube, each tube for one time point, respectively. Meanwhile, the same amount of BSA-Alexa 680 without CRISPRMAX (1.25 μg of BSA-Alexa 680 in 250 μL of solution without CRISPRMAX per tube, each tube for one time point, respectively) was prepared as positive control groups, of which the fluorescence intensity represented the total amount of BSA-Alexa 680 (Vt). All samples of the release groups and control groups were put into a shaking water bath at 37 oC, protecting from the light. For the experimental setup, the solution of CRISPRMAX-BSA-Alexa 680 complexes was added into the upper chamber of the centrifugal filter device (100K, Merck Millipore), followed by centrifugation at 13000 rpm for 5 min; the filtrate of the release groups per setup was collected. The fluorescence intensity of the free BSA-Alexa 680 solution (V0, V1, V2, V4, V8, V12, and V24) in the release groups were measured at each time point. The fluorescence intensity values of the positive control groups (Vt0, Vt1, Vt2, Vt4, Vt8, Vt12, and Vt24) were also measured at each time point. Based on the fluorescence value of the positive controls at each time point, the release percentage was normalized at each time point, and subtracted the proportion of the original unencapsulated free BSA-Alexa 680. For example, the percentage release at the first hour time point was calculated as follows:

Percentage release (%) = [(V1 / Vt1) – (V0 / Vt0)] × 100%

Our results showed that there was no significant releases of the free BSA-Alexa 680 from CRISPRMAX-BSA-Alexa 680 complexes within the first four hour. A small release (< 10%) was detected at the 8th hour, but no further release was detected thereafter (Figure 1e), indicating CRISPRMAX-BSA-Alexa 680 complexes were relatively stable in vitro.

This manuscript is a resubmission of an earlier submission. The following is a list of the peer review reports and author responses from that submission.